# Universal approximation and model compression for radial neural networks

## Abstract

We introduce a class of fully-connected neural networks whose activation functions, rather than being pointwise, rescale feature vectors by a function depending only on their norm. We call such networks *radial neural networks*, extending previous work on rotation equivariant networks that considers rescaling activations in less generality. We prove universal approximation theorems for radial neural networks, including in the more difficult cases of bounded widths and unbounded domains. Our proof techniques are novel, distinct from those in the pointwise case. Additionally, radial neural networks exhibit a rich group of orthogonal change-of-basis symmetries on the vector space of trainable parameters. Factoring out these symmetries leads to a practical lossless model compression algorithm. Optimization of the compressed model by gradient descent is equivalent to projected gradient descent for the full model.

## 1 Introduction

Inspired by biological neural networks, the theory of artificial neural networks has largely focused on pointwise (or "local") nonlinear layers [46, 14], in which the same function $\sigma \colon \mathbb{R} \to \mathbb{R}$ is applied to each coordinate independently:

$$\mathbb{R}^n \to \mathbb{R}^n, \qquad v = (v_1, \ldots, v_n) \mapsto (\sigma(v_1), \sigma(v_2), \ldots, \sigma(v_n)). \tag{1.1}$$

In networks with pointwise nonlinearities, the standard basis vectors in $\mathbb{R}^n$ can be interpreted as "neurons" and the nonlinearity as a "neuron activation." Research has generally focused on finding functions $\sigma$ which lead to more stable training, have less sensitivity to initialization, or are better adapted to certain applications [42, 38, 37, 10, 29]. Many $\sigma$ have been considered, including sigmoid, ReLU, arctangent, ELU, Swish, and others.

However, by setting aside the biological metaphor, it is possible to consider a much broader class of nonlinearities, which are not necessarily pointwise, but instead depend simultaneously on many coordinates. Freedom from the pointwise assumption allows one to design activations that yield expressive function classes with specific advantages. Additionally, certain choices of non-pointwise activations maximize symmetry in the parameter space of the network, leading to compressibility and other desirable properties.

In this paper, we introduce *radial* neural networks which employ non-pointwise nonlinearities called *radial rescaling* activations. Such networks enjoy several provable properties including high model compressibility, symmetry in optimization, and universal approximation. Radial rescaling activations are defined by rescaling each vector by a scalar that

Submitted to 36th Conference on Neural Information Processing Systems (NeurIPS 2022). Do not distribute.

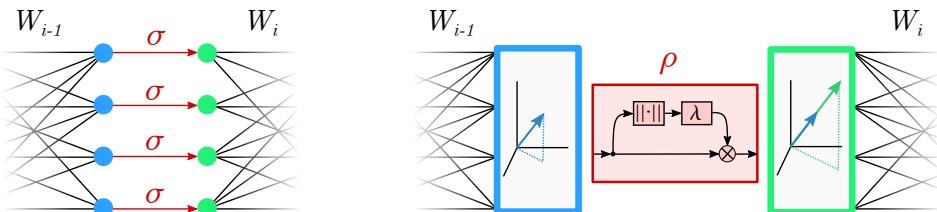

Figure 1: (Left) Pointwise activations distinguish a specific basis of each hidden layer and treat each coordinate independently, see equation 1.1. (Right) Radial rescaling activations rescale each feature vector by a function of the norm, see equation 1.2.

depends only on the norm of the vector:

$$\rho : \mathbb{R}^n \to \mathbb{R}^n, \qquad v \mapsto \lambda(|v|)v, \tag{1.2}$$

where $\lambda$ is a scalar-valued function of the norm. Whereas in the pointwise setting, only the linear layers mix information between different components of the latent features, for radial rescaling, all coordinates of the activation output vector are affected by all coordinates of the activation input vector. The inherent geometric symmetry of radial rescalings makes them particularly useful for designing equivariant neural networks [55, 47, 56, 57].

We note that radial neural networks constitute a simple and previously unconsidered type of multilayer radial basis functions network [4], namely, one where the number of hidden activation neurons (often denoted $N$) in each layer is equal to one. Indeed, pre-composing equation 1.2 with a translation and post-composing with a linear map, one obtains a special case of the local linear model extension of a radial basis functions network.

In our first set of main results, we prove that radial neural networks are in fact *universal approximators*. Specifically, we demonstrate that any asymptotically affine function can be approximated with a radial neural network, suggesting potentially good extrapolation behavior. Moreover, this approximation can be done with bounded width. Our approach to proving these results departs markedly from techniques used in the pointwise case. Additionally, our result is not implied by the universality property of radial basis functions networks in general, and differs in significant ways, particularly in the bounded width property and the approximation of asymptotically affine functions.

In our second set of main results, we exploit parameter space symmetries of radial neural networks to achieve *model compression*. Using the fact that radial rescaling activations commute with orthogonal transformations, we develop a practical algorithm to systematically factor out orthogonal symmetries via iterated QR decompositions. This leads to another radial neural network with fewer neurons in each hidden layer. The resulting model compression algorithm is *lossless*: the compressed network and the original network both have the same value of the loss function on any batch of training data.

Furthermore, we prove that the loss of the compressed model after one step of gradient descent is equal to the loss of the original model after one step of *projected gradient descent*. As explained below, projected gradient descent involves zeroing out certain parameter values after each step of gradient descent. Although training the original network may result in a lower loss function after fewer epochs, in many cases the compressed network takes less time per epoch to train and is faster in reaching a local minimum.

To summarize, our main contributions are:

- A formalization of radial neural networks, a new class of neural networks;

- Universal approximations results for radial neural networks, including: a) approximation of asymptotically affine functions, and b) bounded width approximation;

- Implementation of a lossless compression algorithm for radial neural networks;

- A theorem providing the precise relationship between gradient descent optimization of the original and compressed networks.

## 2 Related work

**Radial rescaling activations.** As noted, radial rescaling activations are a special case of the activations used in radial basis functions networks [4]. Radial rescaling functions have the symmetry property of preserving vector directions, and hence exhibit rotation equivariance. Consequently, examples of such functions, such as the squashing nonlinearity and Norm-ReLU, feature in the study of rotationally equivariant neural networks [55, 47, 56, 57, 26]. However, previous works apply the activation only along the channel dimension, and consider the orthogonal group $O(n)$ only for $n = 2, 3$. In contrast, we consider a radial rescaling activation across the entire hidden layer, and $O(n)$-equivariance where $n$ is the hidden layer dimension. Our constructions echo the vector neurons formalism [15], in which the output of a nonlinearity is a vector rather than a scalar.

**Universal approximation.** Neural networks of arbitrary width and sigmoid activations have long been known to be universal approximators [14]. Universality can also be achieved by bounded width networks with arbitrary depth [36], and generalizes to other activations and architectures [24, 60, 43, 50]. While most work has focused on compact domains, some recent work also considers non-compact domains [28, 54]. The techniques used for pointwise activations do not generalize to radial rescaling activations, where all activation output coordinates are affected by all input coordinates. Consequently, individual radial neural network approximators of two different functions cannot be easily combined to an approximator of the sum of the functions. The standard proof of universal approximation for radial basis functions networks requires an unbounded increase the number of hidden activation neurons, and hence does not apply to the case of radial neural networks [40].

**Groups and symmetry.** Appearances of symmetry in machine learning have generally focused on symmetric input and output spaces. Most prominently, equivariant neural networks incorporate symmetry as an inductive bias and feature weight-sharing constraints based on equivariance. Examples include $G$-convolution, steerable CNN, and Clebsch-Gordon networks [13, 55, 11, 9, 30, 2, 58, 12, 57, 16, 31, 44]. By contrast, our approach to radial neural networks does not depend on symmetries of the input domain, output space, or feedforward mapping. Instead, we exploit parameter space symmetries and thus obtain more general results that apply to domains with no apparent symmetry.

**Model compression.** A major goal in machine learning is to find methods to reduce the number of trainable parameters, decrease memory usage, or accelerate inference and training [8, 61]. Our approach toward this goal differs significantly from most existing methods in that it is based on the inherent symmetry of network parameter spaces. One prior method is *weight pruning*, which removes redundant weights with little loss in accuracy [20, 3, 27]. Pruning can be done during training [18] or at initialization [34, 53]. *Gradient-based pruning* removes weights by estimating the increase in loss resulting from their removal [33, 22, 17, 39]. A complementary approach is *quantization*, which decreases the bit depth of weights [59, 25, 19]. *Knowledge distillation* identifies a small model mimicking the performance of a larger model [5, 23, 1]. *Matrix Factorization* methods replace fully connected layers with lower rank or sparse factored tensors [6, 7, 52, 32, 45, 35] and can often be applied before training. Our method involves a type of matrix factorization based on the QR decomposition; however, rather than aim for rank reduction, we leverage this decomposition to reduce hidden widths via change-of-basis operations on the hidden representations. Close to our method are lossless compression methods which remove stable neurons in ReLU networks [49, 48] or exploit permutation parameter space symmetry to remove neurons [51]; our compression instead follows from the symmetries of the radial rescaling activation. Finally, the compression results of [26], while conceptually similar to ours, are weaker, as (1) the unitary group action is on disjoint layers instead of moving through all layers, and (2) the results are only stated for the squashing nonlinearity.

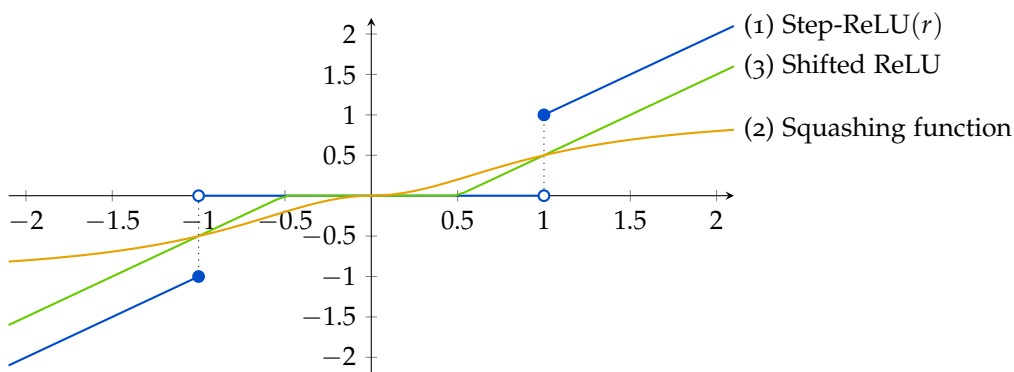

Figure 2: Examples of different radial rescaling functions in $\mathbb{R}^1$, see Example 1.

## 3 Radial neural networks

In this section, we define radial rescaling functions and radial neural networks. Let $h : \mathbb{R} \to \mathbb{R}$ be a function. For any $n \geq 1$, set:

$$h^{(n)} : \mathbb{R}^n \to \mathbb{R}^n \qquad h^{(n)}(v) = h(|v|)\frac{v}{|v|}$$

for $v \neq 0$, and $h^{(n)}(0) = 0$. A function $\rho : \mathbb{R}^n \to \mathbb{R}^n$ is called a *radial rescaling* function if $\rho = h^{(n)}$ for some piecewise differentiable $h : \mathbb{R} \to \mathbb{R}$. Hence, $\rho$ sends each input vector to a scalar multiple of itself, and that scalar depends only on the norm of the vector[1]. It is easy to show that radial rescaling functions commute with orthogonal transformations.

**Example 1.** (1) Step-ReLU, where $h(r) = r$ if $r \geq 1$ and 0 otherwise. In this case, the radial rescaling function is given by

$$\rho : \mathbb{R}^n \to \mathbb{R}^n, \qquad v \mapsto v \text{ if } |v| \geq 1; \qquad v \mapsto 0 \text{ if } |v| < 1 \tag{3.1}$$

(2) The squashing function, where $h(r) = r^2/(r^2 + 1)$. (3) Shifted ReLU, where $h(r) = \max(0, r - b)$ for $r > 0$ and $b$ is a real number. See Figure 2. We refer to [55] and the references therein for more examples and discussion of radial functions.

A *radial neural network* with $L$ layers consists of a positive integer $n_i$ indicating the width of each layer $i = 0, 1, \ldots, L$; the trainable parameters, comprising of a matrix $W_i \in \mathbb{R}^{n_i \times n_{i-1}}$ of weights and a bias vector $b_i \in \mathbb{R}^{n_i}$ for each $i = 1, \ldots, L$; and a radial rescaling function $\rho_i : \mathbb{R}^{n_i} \to \mathbb{R}^{n_i}$ for each $i = 1, \ldots, L$. We refer to the tuple $\mathbf{n} = (n_0, n_1, \ldots, n_L)$ as the *widths vector* of the neural network. The hidden widths vector is $\mathbf{n}^{\text{hid}} = (n_1, n_2, \ldots, n_{L-1})$. The feedforward function $F : \mathbb{R}^{n_0} \to \mathbb{R}^{n_L}$ of a radial neural network is defined in the usual way as an iterated composition of affine maps and activations. Explicitly, set $F_0 = \text{id}_{\mathbb{R}^{n_0}}$ and recursively define the partial feedforward functions for $i = 1, \ldots, L$:

$$F_i : \mathbb{R}^{n_0} \to \mathbb{R}^{n_i}, \qquad x \mapsto \rho_i \left( W_i \circ F_{i-1}(x) + b_i \right)$$

Then the feedforward function is $F = F_L$. Radial neural networks are a special type of radial basis functions network; we explain the connection in Appendix F.

**Remark 2.** If $b_i = 0$ for all $i$, then the feedforward function takes the form $F(x) = W(\mu(x)x)$ where $\mu : \mathbb{R}^n \to \mathbb{R}$ is a scalar-valued function and $W = W_L W_{L-1} \cdots W_1 \in \mathbb{R}^{n_L \times n_0}$ is the product of the weight matrices. If any of the biases are non-zero, then the feedforward function lacks such a simple form.

---

[1]A function $\mathbb{R}^n \to \mathbb{R}$ that depends only on the norm of a vector is known as a *radial* function. Radial rescaling functions rescale each vector according to the radial function $v \mapsto \lambda(|v|) := \frac{h(|v|)}{|v|}$. This explains the connection to Equation 1.2.

## 4 Universal Approximation

In this section, we consider two universal approximation results. The first approximates asymptotically affine functions with a network of unbounded width. The second generalizes to bounded width networks. Proofs appear in Appendix B. Throughout, $B_r(c) = \{x \in \mathbb{R}^n : |x - c| < r\}$ denotes the $r$-ball around a point $c$, and an affine map $\mathbb{R}^n \to \mathbb{R}^m$ is one of the from $L(x) = Ax + b$ for a matrix $A \in \mathbb{R}^{m \times n}$ and $b \in \mathbb{R}^m$.

### 4.1 Approximation of asymptotically affine functions

A continuous function $f : \mathbb{R}^n \to \mathbb{R}^m$ is said to be *asymptotically affine* if there exists an affine map $L : \mathbb{R}^n \to \mathbb{R}^m$ such that, for every $\epsilon > 0$, there is a compact subset $K$ of $\mathbb{R}^n$ such that $|L(x) - f(x)| < \epsilon$ for all $x \in \mathbb{R}^n \setminus K$. In particular, continuous functions with compact support are asymptotically affine. The continuity of $f$ and compactness of $K$ imply that, for any $\epsilon > 0$, there exist $c_1, \ldots, c_N \in K$ and $r_1, \ldots, r_N \in (0, 1)$ such that, first, the union of the balls $B_{r_i}(c_i)$ covers $K$ and, second, for all $i$, we have $f(B_{r_i}(c_i) \cap K) \subseteq B_\epsilon(f(c_i))$. Let $N(f, K, \epsilon)$ be the minimal[2] choice of $N$.

**Theorem 3** (Universal approximation). *Let $f : \mathbb{R}^n \to \mathbb{R}^m$ be an asymptotically affine function. For any $\epsilon > 0$, there exists a compact set $K \subset \mathbb{R}^n$ and a function $F : \mathbb{R}^n \to \mathbb{R}^m$ such that:*

    *1. $F$ is the feedforward function of a radial neural network with $N = N(f, K, \epsilon)$ layers whose hidden widths are $(n + 1, n + 2, \ldots, n + N)$.*

    *2. For any $x \in \mathbb{R}^n$, we have $|F(x) - f(x)| < \epsilon$.*

We note that the approximation in Theorem 3 is valid on all of $\mathbb{R}^n$. To give an idea of the proof, first fix $c_1, \ldots, c_N \in K$ and $r_1, \ldots, r_N \in (0, 1)$ as above. Let $e_1, \ldots, e_N$ be orthonormal basis vectors extending $\mathbb{R}^n$ to $\mathbb{R}^{n+N}$. For $i = 1, \ldots, N$ define affine maps $T_i : \mathbb{R}^{n+i-1} \to \mathbb{R}^{n+i}$ and $S_i : \mathbb{R}^{n+i} \to \mathbb{R}^{n+i}$ by

$$T_i(z) = z - c_i + h_i e_i \qquad S_i(z) = z - (1 + h_i^{-1})\langle e_i, z \rangle e_i + c_i + e_i$$

where $h_i^2 = 1 - r_i^2$ and $\langle e_i, z \rangle$ is the coefficient of $e_i$ in $z$. Setting $\rho_i$ to be Step-ReLU (Equation 3.1) on $\mathbb{R}^{n+i}$, these maps are chosen so that the composition $S_i \circ \rho_i \circ T_i$ maps the points in $B_{r_i}(c_i)$ to $c_i + e_i$, while keeping points outside this ball the same. We now describe a radial neural network with widths $(n, n + 1, \ldots, n + N, m)$ whose feedforward function approximates $f$. For $i = 1, \ldots, N$ the affine map from layer $i - 1$ to layer $i$ is given by $z \mapsto T_i \circ S_{i-1}(z)$, with $S_0 = \mathrm{id}_{\mathbb{R}^n}$. The activation at each hidden layer is Step-ReLU. Let $L$ be the affine map such that $|L - f| < \epsilon$ on $\mathbb{R}^n \setminus K$. The affine map from layer $N$ to the output layer is $\Phi \circ S_N$ where $\Phi : \mathbb{R}^{n+N} \to \mathbb{R}^m$ is the unique affine map determined by $x \mapsto L(x)$ if $x \in \mathbb{R}^n$, and $e_i \mapsto f(c_i) - L(c_i)$. This construction is illustrated in Figure 3.

**Corollary 4.** *Radial neural networks are dense in the space of all continuous functions with respect to the topology of compact convergence, and hence satisfy cc-universality.*

### 4.2 Bounded width approximation

We now turn our attention to a bounded width universal approximation result.

**Theorem 5.** *Let $f : \mathbb{R}^n \to \mathbb{R}^m$ be an asymptotically affine function. For any $\epsilon > 0$, there exists a compact set $K \subset \mathbb{R}^n$ and a function $F : \mathbb{R}^n \to \mathbb{R}^m$ such that:*

    *1. $F$ is the feedforward function of a radial neural network with $N = N(f, K, \epsilon)$ hidden layers whose widths are all $n + m + 1$.*

    *2. For any $x \in \mathbb{R}^n$, we have $|F(x) - f(x)| < \epsilon$.*

The proof, which is more involved than that of Theorem 3, relies on using orthogonal dimensions to represent the domain and the range of $f$, together with an indicator

---

[2]In many cases, the constant $N(f, K, \epsilon)$ can be bounded explicitly. For example, if $K$ is the unit cube in $\mathbb{R}^n$ and $f$ is Lipschitz continuous with Lipschitz constant $R$, then $N(f, K, \epsilon) \leq \left\lceil \frac{R\sqrt{n}}{2\epsilon} \right\rceil^n$.

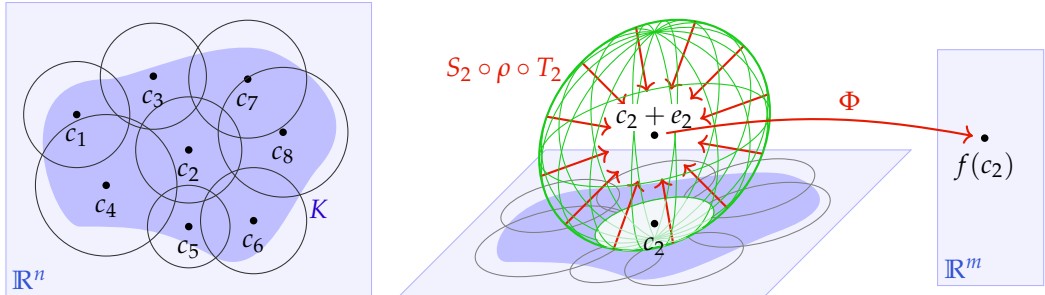

Figure 3: Two layers of the radial neural network used in the proof of Theorem 3. (Left) The compact set $K$ is covered with open balls. (Middle) Points close to $c_2$ (green ball) are mapped to $c_2 + e_2$, all other points are kept the same. (Right) In the final layer, $c_2 + e_2$ is mapped to $f(c_2)$.

dimension to distinguish the two. We regard points in $\mathbb{R}^{n+m+1}$ as triples $(x, y, \theta)$ where $x \in \mathbb{R}^n$, $y \in \mathbb{R}^m$ and $\theta \in \mathbb{R}$. The proof of Theorem 5 parallels that of Theorem 3, but instead of mapping points in $B_{r_i}(c_i)$ to $c_i + e_i$, we map the points in $B_{r_i}((c_i, 0, 0))$ to $(0, \frac{f(c_i) - L(0)}{s}, 1)$, where $s$ is chosen such that different balls do not interfere. The final layer then uses an affine map $(x, y, \theta) \mapsto L(x) + sy$, which takes $(x, 0, 0)$ to $L(x)$, and $(0, \frac{f(c_i) - L(0)}{s}, 1)$ to $f(c_i)$.

We remark on several additional results; see Appendix B for full statements and proofs. The bound of Theorem 5 can be strengthened to $\max(n, m) + 1$ in the case of functions $f : K \to \mathbb{R}^m$ defined on a compact domain $K \subset \mathbb{R}^n$ (i.e., ignoring asymptotic behavior). Furthermore, with more layers, it is possible to reduce that bound to $\max(n, m)$.

## 5 Model compression

In this section, we prove a model compression result. Specifically, we provide an algorithm which, given any radial neural network, computes a different radial neural network with smaller widths. The resulting compressed network has the same feedforward function as the original network, and hence the same value of the loss function on any batch of training data. In other words, our model compression procedure is *lossless*. Although our algorithm is practical and explicit, it reflects more conceptual phenomena, namely, a change-of-basis action on network parameter spaces (Section 5.1).

### 5.1 The parameter space

Suppose a fully connected network has $L$ layers and widths given by the tuple $\mathbf{n} = (n_0, n_1, n_2, \ldots, n_{L-1}, n_L)$. In other words, the $i$-th layer has input width $n_{i-1}$ and output width $n_i$. The parameter space is defined as the vector space of all possible choices of parameter values. Hence, it is given by the following product of vector spaces:

$$\mathsf{Param}(\mathbf{n}) = \left( \mathbb{R}^{n_1 \times n_0} \times \mathbb{R}^{n_2 \times n_1} \times \cdots \times \mathbb{R}^{n_L \times n_{L-1}} \right) \times \left( \mathbb{R}^{n_1} \times \mathbb{R}^{n_2} \times \cdots \times \mathbb{R}^{n_L} \right)$$

We denote an element therein as a pair of tuples $(\mathbf{W}, \mathbf{b})$ where $\mathbf{W} = (W_i \in \mathbb{R}^{n_i \times n_{i-1}})_{i=1}^L$ are the weights and $\mathbf{b} = (b_i \in \mathbb{R}^{n_i})_{i=1}^L$ are the biases. To describe certain symmetries of the parameter space, consider the following product of orthogonal groups, with sizes corresponding to the widths of the hidden layers:

$$O(\mathbf{n}^{\mathrm{hid}}) = O(n_1) \times O(n_2) \times \cdots \times O(n_{L-1})$$

There is a change-of-basis action of $O(\mathbf{n}^{\mathrm{hid}})$ on the parameter space $\mathsf{Param}(\mathbf{n})$. Explicitly, the tuple of orthogonal matrices $\mathbf{Q} = (Q_i)_{i=1}^{L-1} \in O(\mathbf{n}^{\mathrm{hid}})$ transforms the parameter values $(\mathbf{W}, \mathbf{b})$ to $\mathbf{Q} \cdot \mathbf{W} := \left( Q_i W_i Q_{i-1}^{-1} \right)_{i=1}^L$ and $\mathbf{Q} \cdot \mathbf{b} := (Q_i b_i)_{i=1}^L$, where $Q_0 = \mathrm{id}_{n_0}$ and $Q_L = \mathrm{id}_{n_L}$.

## 5.2   Model compression

In order to state the compression result, we first define the reduced widths. Namely, the reduction $\mathbf{n}^{\mathrm{red}} = (n_0^{\mathrm{red}}, n_1^{\mathrm{red}}, \ldots, n_L^{\mathrm{red}})$ of a widths vector $\mathbf{n}$ is defined recursively by setting $n_0^{\mathrm{red}} = n_0$, then $n_i^{\mathrm{red}} = \min(n_i, n_{i-1}^{\mathrm{red}} + 1)$ for $i = 1, \ldots, L-1$, and finally $n_L^{\mathrm{red}} = n_L$. For a tuple $\boldsymbol{\rho} = (\rho_i : \mathbb{R}^{n_i} \to \mathbb{R}^{n_i})_{i=1}^{L}$ of radial rescaling functions, we write $\boldsymbol{\rho}^{\mathrm{red}} = \left( \rho_i^{\mathrm{red}} : \mathbb{R}^{n_i^{\mathrm{red}}} \to \mathbb{R}^{n_i^{\mathrm{red}}} \right)$ for the corresponding tuple of restrictions, which are all radial rescaling functions. The following result relies on Algorithm 1 below.

**Theorem 6.** *Let* $(\mathbf{W}, \mathbf{b}, \boldsymbol{\rho})$ *be a radial neural network with widths* $\mathbf{n}$. *Let* $\mathbf{W}^{\mathrm{red}}$ *and* $\mathbf{b}^{\mathrm{red}}$ *be the weights and biases of the compressed network produced by Algorithm 1. The feedforward function of the original network* $(\mathbf{W}, \mathbf{b}, \boldsymbol{\rho})$ *coincides with that of the compressed network* $(\mathbf{W}^{\mathrm{red}}, \mathbf{b}^{\mathrm{red}}, \boldsymbol{\rho}^{\mathrm{red}})$.

---

**Algorithm 1:** QR Model Compression (`QR-compress`)

---

**input**   : $\mathbf{W}, \mathbf{b} \in \mathsf{Param}(\mathbf{n})$
**output**  : $\mathbf{Q} \in O(\mathbf{n}^{\mathrm{hid}})$ and $\mathbf{W}^{\mathrm{red}}, \mathbf{b}^{\mathrm{red}} \in \mathsf{Param}(\mathbf{n}^{\mathrm{red}})$

$\mathbf{Q}, \mathbf{W}^{\mathrm{red}}, \mathbf{b}^{\mathrm{red}} \leftarrow [\,], [\,], [\,]$                     // initialize output lists
$A_1 \leftarrow [b_1 \quad W_1]$                                // matrix of size $n_1 \times (n_0 + 1)$
**for** $i \leftarrow 1$ **to** $L-1$ **do**                          // iterate through layers
  $\quad Q_i, R_i \leftarrow$ `QR-decomp`$(A_i$ , mode = 'complete')         //  $A_i = Q_i \mathrm{Inc}_i R_i$
  $\quad$ Append $Q_i$ to $\mathbf{Q}$
  $\quad$ Append first column of $R_i$ to $\mathbf{b}^{\mathrm{red}}$          // reduced bias for layer $i$
  $\quad$ Append remainder of $R_i$ to $\mathbf{W}^{\mathrm{red}}$          // reduced weights for layer $i$
  $\quad$ Set $A_{i+1} \leftarrow [b_{i+1} \quad W_{i+1} Q_i \mathrm{Inc}_i]$      // matrix of size $n_{i+1} \times (n_i^{\mathrm{red}} + 1)$
**end**
Append the first column of $A_L$ to $\mathbf{b}^{\mathrm{red}}$          // reduced bias for last layer
Append the remainder of $A_L$ to $\mathbf{W}^{\mathrm{red}}$          // reduced weights for last layer

**return** $\mathbf{Q}$, $\mathbf{W}^{\mathrm{red}}$, $\mathbf{b}^{\mathrm{red}}$

---

We explain the notation of the algorithm. The inclusion matrix $\mathrm{Inc}_i \in \mathbb{R}^{n_i \times n_i^{\mathrm{red}}}$ has ones along the main diagonal and zeros elsewhere. The method `QR-decomp` with mode = 'complete' computes the complete QR decomposition of the $n_i \times (1 + n_{i-1}^{\mathrm{red}})$ matrix $A_i$ as $Q_i \mathrm{Inc}_i R_i$ where $Q_i \in O(n_i)$ and $R_i$ is upper-triangular of size $n_i^{\mathrm{red}} \times (1 + n_{i-1}^{\mathrm{red}})$. The definition of $n_i^{\mathrm{red}}$ implies that either $n_i^{\mathrm{red}} = n_{i-1}^{\mathrm{red}} + 1$ or $n_i^{\mathrm{red}} = n_i$. The matrix $R_i$ is of size $n_i^{\mathrm{red}} \times n_i^{\mathrm{red}}$ in the former case and of size $n_i \times (1 + n_{i-1}^{\mathrm{red}})$ in the latter case.

**Example 7.** Suppose the widths of a radial neural network are $(1, 8, 16, 8, 1)$. Then it has $\sum_{i=1}^{4}(n_{i-1}+1)n_i = 305$ trainable parameters. The reduced network has widths $(1, 2, 3, 4, 1)$ and $\sum_{i=1}^{4}(n_{i-1}^{\mathrm{red}}+1)(n_i^{\mathrm{red}}) = 34$ trainable parameters. Another example appears in Figure 4.

We note that the tuple of matrices $\mathbf{Q}$ produced by Algorithm 1 does not feature in the statement of Theorem 6, but is important in the proof (which appears in Appendix C).

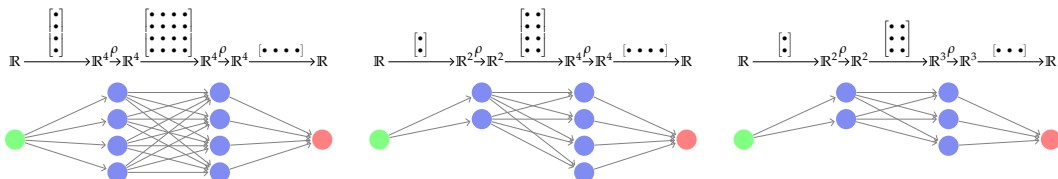

Figure 4: Model compression in 3 steps. Layer widths can be iteratively reduced to 1 greater than the previous. The number of trainable parameters reduces from 33 to 17.

Namely, an induction argument shows that the $i$-th partial feedforward function of the original and reduced models are related via the matrices $Q_i$ and $\text{Inc}_i$. A crucial ingredient in the proof is that radial rescaling activations commute with orthogonal transformations.

## 6 Projected gradient descent

The typical use case for model compression algorithms is to produce a smaller version of the fully trained model which can be deployed to make inference more efficient. It is also worth considering whether compression can be used to accelerate training. For example, for some compression algorithms, the compressed and full models have the same feedforward function after a step of gradient descent is applied to each, and so one can compress before training and still reach the same minimum. Unfortunately, in the context of radial neural networks, compression using Algorithm 1 and then training does not necessarily give the same result as training and then compression (see Appendix D.6 for a counterexample). However, QR-compress does lead to a precise mathematical relationship between optimization of the two models: the loss of the compressed model after one step of gradient descent is equivalent to the loss of (a transformed version of) the original model after one step of projected gradient descent. Proofs appear in Appendix D.

To state our results, fix a tuple of widths $\mathbf{n}$ and a tuple $\boldsymbol{\rho} = (\rho_i : \mathbb{R}^{n_i} \to \mathbb{R}^{n_i})_{i=1}^{L}$ of radial rescaling functions. The loss function $\mathcal{L} : \text{Param}(\mathbf{n}) \to \mathbb{R}$ associated to a batch of training data $\{(x_j, y_j)\} \subseteq \mathbb{R}^{n_0} \times \mathbb{R}^{n_L}$ is defined as taking parameter values $(\mathbf{W}, \mathbf{b})$ to the sum $\sum_j \mathcal{C}(F(x_j), y_j)$ where $\mathcal{C} : \mathbb{R}^{n_L} \times \mathbb{R}^{n_L} \to \mathbb{R}$ is a cost function on the output space, and $F = F_{(\mathbf{W}, \mathbf{b}, \boldsymbol{\rho})}$ is the feedforward of the radial neural network with parameters $(\mathbf{W}, \mathbf{b})$ and activations $\boldsymbol{\rho}$. Similarly, we have a loss function $\mathcal{L}_{\text{red}}$ on the parameter space $\text{Param}(\mathbf{n}^{\text{red}})$ with reduced widths vector. For any learning rate $\eta > 0$, we obtain gradient descent maps:

$$\gamma : \text{Param}(\mathbf{n}) \to \text{Param}(\mathbf{n}) \qquad\qquad \gamma_{\text{red}} : \text{Param}(\mathbf{n}^{\text{red}}) \to \text{Param}(\mathbf{n}^{\text{red}})$$
$$(\mathbf{W}, \mathbf{b}) \mapsto (\mathbf{W}, \mathbf{b}) - \eta \nabla_{(\mathbf{W}, \mathbf{b})} \mathcal{L} \qquad\qquad (\mathbf{V}, \mathbf{c}) \mapsto (\mathbf{V}, \mathbf{c}) - \eta \nabla_{(\mathbf{V}, \mathbf{c})} \mathcal{L}_{\text{red}}$$

We will also consider, for $k \geq 0$, the $k$-fold composition $\gamma^k = \gamma \circ \gamma \circ \cdots \circ \gamma$ and similarly for $\gamma_{\text{red}}$. The *projected gradient descent* map on $\text{Param}(\mathbf{n})$ is given by:

$$\gamma_{\text{proj}} : \text{Param}(\mathbf{n}) \to \text{Param}(\mathbf{n}), \qquad (\mathbf{W}, \mathbf{b}) \mapsto \text{Proj}\left(\gamma(\mathbf{W}, \mathbf{b})\right)$$

where the map Proj zeroes out all entries in the bottom left $(n_i - n_i^{\text{red}}) \times n_{i-1}^{\text{red}}$ submatrix of $W_i - \nabla_{W_i} \mathcal{L}$, and the bottom $(n_i - n_i^{\text{red}})$ entries in $b_i - \nabla_{b_i} \mathcal{L}$, for each $i$. Schematically:

$$W_i - \nabla_{W_i} \mathcal{L} = \begin{bmatrix} * & * \\ * & * \end{bmatrix} \mapsto \begin{bmatrix} * & * \\ 0 & * \end{bmatrix}, \qquad b_i - \nabla_{b_i} \mathcal{L} = \begin{bmatrix} * \\ * \end{bmatrix} \mapsto \begin{bmatrix} * \\ 0 \end{bmatrix}$$

To state the following theorem, let $\mathbf{W}^{\text{red}}, \mathbf{b}^{\text{red}}, \mathbf{Q} = \text{QR-compress}(\mathbf{W}, \mathbf{b})$ be the outputs of Algorithm 1 applied to $(\mathbf{W}, \mathbf{b}) \in \text{Param}(\mathbf{n})$. Hence $(\mathbf{W}^{\text{red}}, \mathbf{b}^{\text{red}}) \in \text{Param}(\mathbf{n}^{\text{red}})$ are the parameters of the compressed model, and $\mathbf{Q} \in O(\mathbf{n}^{\text{hid}})$ is an orthogonal parameter symmetry. We also consider the action (Section 5.1) of $\mathbf{Q}^{-1}$ applied to $(\mathbf{W}, \mathbf{b})$.

**Theorem 8.** *Let* $\mathbf{W}^{\text{red}}, \mathbf{b}^{\text{red}}, \mathbf{Q} = \text{QR-compress}(\mathbf{W}, \mathbf{b})$ *be the outputs of Algorithm 1 applied to* $(\mathbf{W}, \mathbf{b}) \in \text{Param}(\mathbf{n})$. *Set* $\mathbf{U} = \mathbf{Q}^{-1} \cdot (\mathbf{W}, \mathbf{b}) - (\mathbf{W}^{\text{red}}, \mathbf{b}^{\text{red}})$. *For any* $k \geq 0$, *we have:*

$$\gamma^k(\mathbf{W}, \mathbf{b}) = \mathbf{Q} \cdot \gamma^k(\mathbf{Q}^{-1} \cdot (\mathbf{W}, \mathbf{b})) \qquad\qquad \gamma_{\text{proj}}^k(\mathbf{Q}^{-1} \cdot (\mathbf{W}, \mathbf{b})) = \gamma_{\text{red}}^k(\mathbf{W}^{\text{red}}, \mathbf{b}^{\text{red}}) + \mathbf{U}.$$

We conclude that gradient descent with initial values $(\mathbf{W}, \mathbf{b})$ is equivalent to gradient descent with initial values $\mathbf{Q}^{-1} \cdot (\mathbf{W}, \mathbf{b})$ since at any stage we can apply $\mathbf{Q}^{\pm 1}$ to move from one to the other. Furthermore, projected gradient descent with initial values $\mathbf{Q}^{-1} \cdot (\mathbf{W}, \mathbf{b})$ is equivalent to gradient descent on $\text{Param}(\mathbf{n}^{\text{red}})$ with initial values $(\mathbf{W}^{\text{red}}, \mathbf{b}^{\text{red}})$ since at any stage we can move from one to the other by $\pm \mathbf{U}$. Neither $\mathbf{Q}$ nor $\mathbf{U}$ depends on $k$.

## 7  Experiments

In addition to the theoretical results in this work, we provide an implementation of Algorithm 1, in order to validate the claims of Theorems 6 and 8 empirically, as well as to quantify real-world performance. Full experimental details are in Appendix E.

**(1) Empirical verification of Theorem 6.** We learn the function $f(x) = e^{-x^2}$ from samples using a radial neural network with widths $\mathbf{n} = (1, 6, 7, 1)$ and activation the radial shifted sigmoid $h(x) = 1/(1 + e^{-x+s})$. Applying QR-compress gives a compressed radial neural network with widths $\mathbf{n}^{\text{red}} = (1, 2, 3, 1)$. Theorem 6 implies that the respective neural functions $F$ and $F_{\text{red}}$ are equal. Over 10 random initializations, the mean absolute error is negligible up to machine precision: $(1/N) \sum_j |F(x_j) - F_{\text{red}}(x_j)| = 1.31 \cdot 10^{-8} \pm 4.45 \cdot 10^{-9}$.

**(2) Empirical verification of Theorem 8.** The claim is that training the transformed model with parameters $\mathbf{Q}^{-1} \cdot (\mathbf{W}, \mathbf{b})$ and objective $\mathcal{L}$ by projected gradient descent coincides with training the reduced model with parameters $(\mathbf{W}^{\text{red}}, \mathbf{b}^{\text{red}})$ and objective $\mathcal{L}_{\text{red}}$ by usual gradient descent. We verified this on synthetic data as above. Over 10 random initializations, the loss functions after training match: $|\mathcal{L} - \mathcal{L}_{\text{red}}| = 4.02 \cdot 10^{-9} \pm 7.01 \cdot 10^{-9}$.

**(3) The compressed model trains faster.** Our compression method may be applied before training to produce a smaller model class which *trains* faster without sacrificing accuracy. We demonstrate this in learning the function $f : \mathbb{R}^2 \to \mathbb{R}^2$ sending $(t_1, t_2)$ to $(e^{-t_1^2}, e^{-t_2^2})$ using a radial neural network with widths $\mathbf{n} = (2, 16, 64, 128, 16, 2)$ and activation the radial sigmoid $h(r) = 1/(1 + e^{-r})$. Applying QR-compress gives a compressed network with widths $\mathbf{n}^{\text{red}} = (2, 3, 4, 5, 6, 2)$. We trained both models until the training loss was $\leq 0.01$. Over 10 random initializations on our system, the reduced network trained in $15.32 \pm 2.53$ seconds and the original network trained in $31.24 \pm 4.55$ seconds.

## 8  Conclusions and Discussion

This paper demonstrates that radial neural networks are universal approximators and that their parameter spaces exhibit a rich symmetry group, leading to a model compression algorithm. The results of this work combine to build a theoretical foundation for the use of radial neural networks, and suggest that radial neural networks hold promise for wider practical applicability. Furthermore, this work makes an argument for considering the advantages of non-pointwise nonlinearities in neural networks.

There are two main limitations of our results, each providing an opportunity for future work. First, our universal approximation constructions currently work only for Step-ReLU radial rescaling radial activations; it would be desirable to generalize to other activations. Additionally, Theorem 6 achieves compression only for networks whose widths satisfy $n_i > n_{i-1} + 1$ for some $i$. Neural networks which do not have increasing widths anywhere in their architecture, such as encoders, would not be compressible.

Further extensions of this work include: First, little is currently known about the stability properties of radial neural networks during training, as well as their sensitivity to initialization. Second, radial rescaling activations provide an extreme case of symmetry; there may be benefits to combining radial and pointwise activations within a single network, for example, through 'block' radial rescaling functions. Our techniques may yield weaker compression properties for more general radial basis functions networks; radial neural networks may be the most compressible such networks. Third, the parameter space symmetries may provide a key ingredient in analyzing the gradient flow dynamics of radial neural networks and computation of conserved quantities. Fourth, radial rescaling activations can be used within convolutional or group-equivariant NNs. Finally, based on the theoretical advantages laid out in this paper, future work will explore empirically applications in which we expect radial networks to outperform alternate methods. Such potential applications include data spaces with circular or distance-based class boundaries.

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

# A  Organization of the appendices

This paper is a contribution to the mathematical foundations of machine learning, and our results are motivated by expanding the applicability and performance of neural networks. At the same time, we give precise mathematical formulations of our results and proofs. The purposes of these appendices are several:

1. To clarify the mathematical conventions and terminology, thus making the paper more accessible.

2. To provide full proofs of the main results.

3. To develop context around various construction appearing in the main text.

4. To discuss in detail examples, special cases, and generalizations of our results.

We now give a summary of the contents of the appendices.

Appendix B contains proofs the universal approximation results (Theorems 3 and 5) stated in Section 4 of the main text, as well as proofs of additional bounded width results. The proofs use notation given in Appendix B.1, and rely on preliminary topological considerations given in Appendix B.2.

In Appendix C, we give a proof of the model compression result given in Theorem 6, which appears in Section 5. For clarity and background we begin the appendix with a discussion of the version of the QR decomposition relevant for our purposes (Appendix C.1). We also establish elementary properties of radial rescaling activations (Appendix C.2).

The focus of Appendix D is projected gradient descent, elaborating on Section 6. We first prove a result on the interaction of gradient descent and orthogonal transformations (Appendix D.1), before formulating projected gradient descent in more detail (Appendix D.2), and introducing the so-called interpolating space (Appendix D.3). We restate Theorem 8 in more convenient notation (Appendix D.4) before proceeding to the proof (Appendix D.5).

Appendix E contains implementation details for the experiments summarized in Section 7. Our implementations use shifted radial rescaling activations, which we formulate in Appendix E.1.

Appendix F explains the connection between our constructions and radial basis functions networks. While radial neural networks turn out to be a specific type of radial basis functions network, our universality results are not implied by those for general radial basis functions networks.

# B  Universal approximation proofs and additional results

In this section, we provide full proofs of the universal approximation (UA) results for radial neural networks, as stated in Section 4. In order to do so, we first clarify our notational conventions (Appendix B.1), and collect basic topological results (Appendix B.2).

## B.1  Notation

Recall that, for a point $c$ in the Euclidean space $\mathbb{R}^n$ and a positive real number $r$, we denote the $r$-ball around $c$ by $B_r(c) = \{x \in \mathbb{R}^n \mid |x - c| < r\}$. All networks in this section have the Step-ReLU radial rescaling activation function, defined as:

$$\rho : \mathbb{R}^n \longrightarrow \mathbb{R}^n, \qquad z \longmapsto \begin{cases} z & \text{if } |z| \geq 1 \\ 0 & \text{otherwise} \end{cases}$$

Throughout, $\circ$ denotes the composition of functions. We identify a linear map with a corresponding matrix (in the standard bases). In the case of linear maps, the operation $\circ$

can be be identified with matrix multiplication. Recall also that an affine map $L : \mathbb{R}^n \to \mathbb{R}^m$ is one of the from $L(x) = Ax + b$ for a matrix $A \in \mathbb{R}^{m \times n}$ and $b \in \mathbb{R}^m$.

## B.2 Topology

Let $K$ be a compact subset of $\mathbb{R}^n$ and let $f : K \to \mathbb{R}^m$ be a continuous function.

**Lemma 9.** *For any $\epsilon > 0$, there exist $c_1, \ldots, c_N \in K$ and $r_1, \ldots, r_N \in (0, 1)$ such that, first, the union of the balls $B_{r_i}(c_i)$ covers $K$; second, for all $i$, we have $f(B_{r_i}(c_i) \cap K) \subseteq B_\epsilon(f(c_i))$.*

*Proof.* The continuity of $f$ implies that for each $c \in K$, there exists $r = r_c$ such that $f(B_{r_c}(c) \cap K) \subseteq B_\epsilon(f(c))$. The subsets $B_{r_c}(c) \cap K$ form an open cover of $K$. The compactness of $K$ implies that there is a finite subcover. The result follows. $\square$

We also prove a variation of Lemma 9 that additionally guarantees that none of the balls in the cover of $K$ contains the center point of another ball.

**Lemma 10.** *For any $\epsilon > 0$, there exist $c_1, \ldots, c_M \in K$ and $r_1, \ldots, r_M \in (0, 1)$ such that, first, the union of the balls $B_{r_i}(c_i)$ covers $K$; second, for all $i$, we have $f(B_{r_i}(c_i)) \subseteq B_\epsilon(f(c_i))$; and, third, $|c_i - c_j| \geq r_i$.*

*Proof.* Because $f$ is continuous on a compact domain, it is uniformly continuous. So, there exists $r > 0$ such that $f(B_r(c) \cap K) \subseteq B_\epsilon(f(c))$ for each $c \in K$. Because $K$ is compact it has a finite volume, and so does $B_{r/2}(K) = \bigcup_{c \in K} B_{r/2}(c)$. Hence, there exists a finite maximal packing of $B_{r/2}(K)$ with balls of radius $r/2$. That is, a collection $c_1, \ldots, c_M \in B_{r/2}(K)$ such that, for all $i$, $B_{r/2}(c_i) \subseteq B_{r/2}(K)$ and, for all $j \neq i$, $B_{r/2}(c_i) \cap B_{r/2}(c_j) = \emptyset$. The first condition implies that $c_i \in K$. The second condition implies that $|c_i - c_j| \geq r$. Finally, we argue that $K \subseteq \bigcup_{i=1}^M B_r(c_i)$. To see this, suppose, for a contradiction, that $x \in K$ does not belong to $\bigcup_{i=1}^M B_r(c_i)$. Then $B_{r/2}(c_i) \cap B_{r/2}(x) = \emptyset$, and $x$ could be added to the packing, which contradicts the fact that the packing was chosen to be maximal. So the union of the balls $B_r(c_i)$ covers $K$. $\square$

We turn our attention to the minimal choices of $N$ and $M$ in Lemmas 9 and 10.

**Definition 11.** Given $f : K \to \mathbb{R}^m$ continuous and $\epsilon > 0$, let $N(f, K, \epsilon)$ be the minimal choice of $N$ in Lemma 9, and let $M(f, K, \epsilon)$ be the minimal choice of $M$ in Lemma 10.

Observe that $M(f, K, \epsilon) \geq N(f, K, \epsilon)$. In many cases, it is possible to give explicit bounds for the constants $N(f, K, \epsilon)$ and $M(f, K, \epsilon)$. As an illustration, we give the argument in the case that $K$ is the closed unit cube in $\mathbb{R}^n$ and $f : K \to \mathbb{R}^m$ is Lipschtiz continuous.

**Proposition 12.** *Let $K = [0, 1]^n \subset \mathbb{R}^n$ be the (closed) unit cube and let $f : K \to \mathbb{R}^m$ be Lipschitz continuous with Lipschitz constant $R$. For any $\epsilon > 0$, we have:*

$$N(f, K, \epsilon) \leq \left\lceil \frac{R\sqrt{n}}{2\epsilon} \right\rceil^n \quad \text{and} \quad M(f, K, \epsilon) \leq \frac{\Gamma(n/2 + 1)}{\pi^{n/2}} \left( 2 + \frac{2R}{\epsilon} \right)^n.$$

*Proof.* For the first inequality, observe that the unit cube can be covered with $\left\lceil \frac{R\sqrt{n}}{2\epsilon} \right\rceil^n$ cubes of side length $\frac{2\epsilon}{R\sqrt{n}}$. Each cube is contained in a ball of radius $\frac{\epsilon}{R}$ centered at the center of the cube. (In general, a cube of side length $a$ in $\mathbb{R}^n$ is contained in a ball of radius $\frac{a\sqrt{n}}{2}$.) Lipschitz continuity implies that, for all $x, x' \in K$, if $|x - x'| < \epsilon/R$ then $|f(x) - f(x')| \leq R|x - x'| < \epsilon$.

For the second inequality, let $r = \epsilon/R$. Lipschitz continuity implies that, for all $x, x' \in K$, if $|x - x'| < r$ then $|f(x) - f(x')| \leq R|x - x'| < \epsilon$. The $n$-dimensional volume of the set of points with distance at most $r/2$ to the unit cube is $\text{vol}(B_{r/2}(K)) \leq (1 + r)^n$. The volume

of a ball with radius $r/2$ is $\text{vol}(B_{r/2}(0)) = \frac{\pi^{n/2}}{\Gamma(n/2+1)}(r/2)^n$. Hence, any packing of $B_{r/2}(K)$ with balls of radius $r/2$ consists of at most

$$\frac{\text{vol}(B_{r/2}(K))}{\text{vol}(B_{r/2}(0))} \leq \frac{\Gamma(n/2+1)}{\pi^{n/2}}\left(2+\frac{2R}{\epsilon}\right)^n$$

such balls. So there also exists a maximal packing with at most that many balls. This packing can be used in the proof of Lemma 10, which implies that it is a bound on $M(f,K,\epsilon)$. $\qquad\square$

We note in passing that any differentiable function $f : K \to \mathbb{R}^n$ on a compact subset $K$ of $\mathbb{R}^n$ is Lipschitz continuous. Indeed, the compactness of $K$ implies that there exists $R$ such that $|f'(x)| \leq R$ for all $x \in K$. Then one can take $R$ to be the Lipschitz constant of $f$.

### B.3 Proof of Theorem 3: UA for asymptotically affine functions

In this section, we restate and prove Theorem 3, which proves that radial neural networks are universal approximators of asymptotically affine functions. We recall the definition of such functions:

**Definition 13.** A function $f : \mathbb{R}^n \to \mathbb{R}^m$ is *asymptotically affine* if there exists an affine function $L : \mathbb{R}^n \to \mathbb{R}^m$ such that, for all $\epsilon > 0$, there exists a compact set $K \subset \mathbb{R}^n$ such that $|L(x) - f(x)| < \epsilon$ for all $x \in \mathbb{R}^n \setminus K$. We say that $L$ is the limit of $f$.

**Remark 14.** An *asymptotically linear* function is defined in the same way, except $L$ is taken to be linear (i.e., given just by applying matrix multiplication without translation). Hence any asymptotically linear function is in particular an asymptotically affine function, and Theorem 3 applies to asymptotically linear functions as well.

Given an asymptotically affine function $f : \mathbb{R}^n \to \mathbb{R}^m$ and $\epsilon > 0$, let $K$ be a compact set as in Definition 13. We apply Lemma 9 to the restriction $f|_K$ of $f$ to $K$ and produce a minimal constant $N = N(f|_K, K, \epsilon)$ as in Definition 11. We write simply $N(f, K, \epsilon)$ for this constant.

**Theorem 3** (Universal approximation). *Let $f : \mathbb{R}^n \to \mathbb{R}^m$ be an asymptotically affine function. For any $\epsilon > 0$, there exists a compact set $K \subset \mathbb{R}^n$ and a function $F : \mathbb{R}^n \to \mathbb{R}^m$ such that:*

  *1. $F$ is the feedforward function of a radial neural network with $N = N(f, K, \epsilon)$ layers whose hidden widths are $(n+1, n+2, \ldots, n+N)$.*

  *2. For any $x \in \mathbb{R}^n$, we have $|F(x) - f(x)| < \epsilon$.*

*Proof.* By the hypothesis on $f$, there exists an affine function $L : \mathbb{R}^n \to \mathbb{R}^m$ and a compact set $K \subset \mathbb{R}^n$ such that $|L(x) - f(x)| < \epsilon$ for all $x \in \mathbb{R}^n \setminus K$. Abbreviate $N(f, K, \epsilon)$ by $N$. As in Lemma 9, fix $c_1, \ldots, c_N \in K$ and $r_1, \ldots, r_N \in (0,1)$ such that, first, the union of the balls $B_{r_i}(c_i)$ covers $K$ and, second, for all $i$, we have $f(B_{r_i}(c_i)) \subseteq B_\epsilon(f(c_i))$. Let $U = \bigcup_{i=1}^N B_{r_i}(c_i)$, so that $K \subset U$. Define $F : \mathbb{R}^n \to \mathbb{R}^m$ as:

$$F(x) = \begin{cases} L(x) & \text{if } x \notin U \\ f(c_j) & \text{where } j \text{ is the smallest index with } x \in B_{r_j}(c_j) \end{cases}$$

If $x \notin U$, then $|F(x) - f(x)| = |L(x) - f(x)| < \epsilon$. Hence suppose $x \in U$. Let $j$ be the smallest index such that $x \in B_{r_j}(c_j)$. Then $F(x) = f(c_j)$, and, by the choice of $r_j$, we have:

$$|F(x) - f(x)| = |f(c_j) - f(x)| < \epsilon.$$

We proceed to show that $F$ is the feedforward function of a radial neural network. Let $e_1, \ldots, e_N$ be orthonormal basis vectors extending $\mathbb{R}^n$ to $\mathbb{R}^{n+N}$. We regard each $\mathbb{R}^{n+i-1}$ as a subspace of $\mathbb{R}^{n+i}$ by embedding into the first $n+i-1$ coordinates. For $i = 1, \ldots, N$, we set $h_i = \sqrt{1 - r_i^2}$ and define the following affine transformations:

$$T_i : \mathbb{R}^{n+i-1} \to \mathbb{R}^{n+i} \qquad\qquad S_i : \mathbb{R}^{n+i} \to \mathbb{R}^{n+i}$$

$$z \mapsto z - c_i + h_i e_i \qquad\qquad z \mapsto z - (1 + h_i^{-1})\langle e_i, z \rangle e_i + c_i + e_i$$

where $\langle e_i, z \rangle$ is the coefficient of $e_i$ in $z$. Consider the radial neural network with widths $(n, n+1, \ldots, n+N, m)$, whose affine transformations and activations are given by:

- For $i = 1, \ldots, N$ the affine transformation from layer $i-1$ to layer $i$ is given by $z \mapsto T_i \circ S_{i-1}(z)$, where $S_0 = \mathrm{id}_{\mathbb{R}^n}$.

- The activation function at the $i$-th hidden layer is Step-ReLU on $\mathbb{R}^{n+i}$, that is:

$$\rho_i : \mathbb{R}^{n+i} \longrightarrow \mathbb{R}^{n+i}, \qquad z \longmapsto \begin{cases} z & \text{if } |z| \geq 1 \\ 0 & \text{otherwise} \end{cases}$$

- The affine transformation from layer $i = N$ to the output layer is

$$z \mapsto \Phi_{L,f,\mathbf{c}} \circ S_N(z)$$

where $\Phi_{L,f,\mathbf{c}}$ is the affine transformation given by:

$$\Phi_{L,f,\mathbf{c}} : \mathbb{R}^{n+N} \to \mathbb{R}^m, \qquad x + \sum_{i=1}^{N} a_i e_i \; \mapsto \; L(x) + \sum_{i=1}^{N} a_i(f(c_i) - L(c_i))$$

which can be shown to be affine when $L$ is affine. Indeed, write $L(x) = Ax + b$ where $A$ is a matrix in $\mathbb{R}^{m \times n}$ and $b \in R^m$ is a vector. Then $\Phi_{L,f,\mathbf{c}}$ is the composition of the linear map given by the matrix

$$[A \quad f(c_1) - L(c_1) \quad f(c_2) - L(c_2) \quad \cdots \quad f(c_N) - L(c_N)] \in \mathbb{R}^{m \times (n+N)}$$

and translation by $b \in \mathbb{R}^m$. Note that we regard each $f(c_i) - L(c_i) \in \mathbb{R}^m$ as a column vector in the matrix above.

We claim that the feedforward function of the above radial neural network is exactly $F$. To show this, we first state a lemma, whose (omitted) proof is an elementary computation.

**Lemma 3.1.** For $i = 1, \ldots, N$, the composition $S_i \circ T_i$ is the embedding $\mathbb{R}^{n+i-1} \hookrightarrow \mathbb{R}^{n+i}$.

Next, recursively define $G_i : \mathbb{R}^n \to \mathbb{R}^{n+i}$ via

$$G_i = S_i \circ \rho_i \circ T_i \circ G_{i-1},$$

where $G_0 = \mathrm{id}_{\mathbb{R}^n}$. The function $G_i$ admits an direct formulation:

**Proposition 3.2.** For $i = 0, 1, \ldots, N$, we have:

$$G_i(x) = \begin{cases} x & \text{if } x \notin \bigcup_{j=1}^{i} B_{r_j}(c_j) \\ c_j + e_j & \text{where } j \leq i \text{ is the smallest index with } x \in B_{r_j}(c_j) \end{cases}.$$

*Proof.* We proceed by induction. The base step $i = 0$ is immediate. For the induction step, assume the claim is true for $i-1$, where $0 \leq i-1 < N$. There are three cases to consider.

**Case 1**. Suppose $x \notin \bigcup_{j=1}^{i} B_{r_j}(c_j)$. Then in particular $x \notin \bigcup_{j=1}^{i-1} B_{r_j}(c_j)$, so the induction hypothesis implies that $G_{i-1}(x) = x$. Additionally, $x \notin B_{r_i}(c_i)$, so:

$$|T_i(x)| = |x - c_i + h_i e_i| = \sqrt{|x - c_i| + h_i^2} \geq \sqrt{r_i^2 + 1 - r_i^2} = 1.$$

Using the definition of $\rho_i$ and Lemma 3.1, we compute:

$$G_i(x) = S_i \circ \rho_i \circ T_i \circ G_{i-1}(x) = S_i \circ \rho_i \circ T_i(x) = S_i \circ T_i(x) = x.$$

**Case 2**. Suppose $x \in B_j \setminus \bigcup_{k=1}^{j-1} B_{r_k}(c_k)$ for some $j \leq i - 1$. Then the induction hypothesis implies that $G_{i-1}(x) = c_j + e_j$. We compute:

$$|T_i(c_j + e_j)| = |c_j + e_j - c_i + h_i e_i| > |e_j| = 1.$$

Therefore,

$$G_i(x) = S_i \circ \rho_i \circ T_i(c_j + e_j) = S_i \circ T_i(c_j + e_j) = c_j + e_j.$$

**Case 3.** Finally, suppose $x \in B_i \setminus \bigcup_{j=1}^{i-1} B_{r_j}(c_j)$. The induction hypothesis implies that $G_{i-1}(x) = x$. Since $x \in B_{r_i}(c_i)$, we have:

$$|T_i(x)| = |x - c_i + h_i e_i| = \sqrt{|x - c_i| + h_i^2} < \sqrt{r_i^2 + 1 - r_i^2} = 1.$$

Therefore:

$$G_i(x) = S_i \circ \rho_i \circ T_i(x) = S_i(0) = c_i + e_i.$$

This completes the proof of the proposition. $\qquad\square$

Finally, we show that the function $F$ defined at the beginning of the proof is the feedforward function of the above radial neural network. The computation is elementary:

$$\begin{aligned}
F_{\text{feedforward}} &= \Phi_{L,f,\mathbf{c}} \circ S_N \circ \rho_N \circ T_N \circ S_{N-1} \circ \rho_{N-1} \circ T_{N-1} \circ \cdots S_1 \circ \rho_1 \circ T_1 \\
&= \Phi_{L,f,\mathbf{c}} \circ G_N \\
&= F
\end{aligned}$$

where the first equality follows from the definition of the feedforward function, the second from the definition of $G_N$, and the last from the case $i = N$ of Proposition 3.2 together with the definition of $\Phi_{L,f,\mathbf{c}}$. This completes the proof of the theorem. $\qquad\square$

## B.4 Proof of Theorem 5: bounded width UA for asymptotically affine functions

We restate and prove Theorem 5, which strengthens Theorem 3 by providing a bounded width radial neural network approximation of any asymptotically affine function.

**Theorem 5.** *Let $f : \mathbb{R}^n \to \mathbb{R}^m$ be an asymptotically affine function. For any $\epsilon > 0$, there exists a compact set $K \subset \mathbb{R}^n$ and a function $F : \mathbb{R}^n \to \mathbb{R}^m$ such that:*

*1. $F$ is the feedforward function of a radial neural network with $N = N(f, K, \epsilon)$ hidden layers whose widths are all $n + m + 1$.*

*2. For any $x \in \mathbb{R}^n$, we have $|F(x) - f(x)| < \epsilon$.*

*Proof.* By the hypothesis on $f$, there exists an affine function $L : \mathbb{R}^n \to \mathbb{R}^m$ and a compact set $K \subset \mathbb{R}^n$ such that $|L(x) - f(x)| < \epsilon$ for all $x \in \mathbb{R}^n \setminus K$. Given $\epsilon > 0$, let $N = N(f, K, \epsilon)$ and use Lemma 9 to choose $c_1, \ldots, c_N \in K$ and $r_1, \ldots, r_N \in (0, 1)$ such that the union of the balls $B_{r_i}(c_i)$ covers $K$, and, for all $i$, we have $f(B_{r_i}(c_i)) \subseteq B_\epsilon(f(c_i))$. Let $s$ be the minimal non-zero value of $|f(c_i) - f(c_j)|$ for $i, j \in \{1, \ldots, N\}$, that is, $s = \min_{i,j, f(c_i) \neq f(c_j)} |f(c_i) - f(c_j)|$.

Using the decomposition $\mathbb{R}^{n+m+1} \cong \mathbb{R}^n \times \mathbb{R}^m \times \mathbb{R}$, we write elements of $\mathbb{R}^{n+m+1}$ as $(x, y, \theta)$, where $x \in \mathbb{R}^n$, $y \in \mathbb{R}^m$, and $\theta \in \mathbb{R}$. For $i = 1, \ldots, N$, set:

$$T_i : \mathbb{R}^{n+m+1} \to \mathbb{R}^{n+m+1}, \qquad (x, y, \theta) \mapsto \left( x - (1 - \theta)c_i \,,\, y - \theta\frac{f(c_i) - L(0)}{s} \,,\, (1 - \theta)h_i \right)$$

where $h_i = \sqrt{1 - r_i^2}$. Note that $T_i$ is an invertible affine transformation, whose inverse is given by:

$$T_i^{-1}(x, y, \theta) = \left( x + \frac{\theta}{h_i}c_i \,,\, y + \left(1 - \frac{\theta}{h_i}\right)\frac{f(c_i) - L(0)}{s} \,,\, 1 - \frac{\theta}{h_i} \right)$$

For $i = 1, \ldots, N$, define $G_i : \mathbb{R}^n \to \mathbb{R}^{n+m+1}$ via the following recursive definition:

$$G_i = T_i^{-1} \circ \rho \circ T_i \circ G_{i-1},$$

where $G_0(x) = (x, 0, 0) : \mathbb{R}^n \hookrightarrow \mathbb{R}^{n+m+1}$ is the inclusion, and $\rho : \mathbb{R}^{n+m+1} \to \mathbb{R}^{n+m+1}$ is Step-ReLU on $\mathbb{R}^{n+m+1}$. We claim that, for $x \in \mathbb{R}^n$, we have:

$$G_i(x) = \begin{cases} (x, 0, 0) & \text{if } x \notin \bigcup_{j=1}^i B_{r_j}(c_j) \\ \left(0, \frac{f(c_j) - L(0)}{s}, 1\right) & \text{where } j \leq i \text{ is the smallest index with } x \in B_{r_j}(c_j) \end{cases}$$

This claim can be verified by a straightforward induction argument, similar to the one given in the proof of Proposition 3.2, and using the following key facts:

- For $x \in \mathbb{R}^n$, $\left|T_i\left((x, 0, 0)\right)\right| = \left|(x - c_i, 0, h_i)\right| < 1$ if and only if $|x - c_i| < r_i$.

- $T_i^{-1}(0) = \left(0, \frac{f(c_i) - L(0)}{s}, 1\right)$.

- $T_i\left(\left(0, \frac{f(c_j) - L(0)}{s}, 1\right)\right) = \left(0, \frac{f(c_j) - f(c_i)}{s}, 0\right)$, which, by the choice of $s$, has norm at least 1 if $f(c_j) \neq f(c_i)$, and is 0 if $f(c_j) = f(c_i)$.

Let $\Phi : \mathbb{R}^{n+m+1} \to \mathbb{R}^m$ denote the affine map sending $(x, y, \theta)$ to $L(x) + sy$. It follows that $F = \Phi \circ G_N$ satisfies

$$F(x) = \begin{cases} L(x) & \text{if } x \notin \bigcup_{j=1}^N B_{r_j}(c_j) \\ f(c_j) & \text{where } j \text{ is the smallest index with } x \in B_{r_j}(c_j) \end{cases}$$

By construction, $F$ is the feedforward function of a radial neural network with $N$ hidden layers whose widths are all $n + m + 1$. Let $x \in \mathbb{R}^n$. If $x \in K$, let $j$ be the smallest index such that $x \in B_{r_j}(c_j)$. Then $F(x) = f(c_j)$, and, by the choice of $r_j$, we have $|F(x) - f(x)| = |f(c_j) - f(x)| < \epsilon$. Otherwise, $x \in \mathbb{R}^n \setminus K$, and $|F(x) - f(x)| = |L(x) - f(x)| < \epsilon$. $\square$

## B.5 Additional result: bound of $\max(n, m) + 1$

We state and prove an additional bounded width result. In contrast to the results above, the theorem below only holds for functions defined on a compact domain, without assumptions about the asymptotic behavior. The proof is an adaptation of the proof of Theorem 5, so we give only a sketch.

**Theorem 15.** *Let $f : K \to \mathbb{R}^m$ be a continuous function, where $K$ is a compact subset of $\mathbb{R}^n$. For any $\epsilon > 0$, there exists $F : \mathbb{R}^n \to \mathbb{R}^m$ such that:*

1. *$F$ is the feedforward function of a radial neural network with $N(f, K, \epsilon)$ hidden layers whose widths are all $\max(n, m) + 1$.*

2. *For any $x \in K$, we have $|F(x) - f(x)| < \epsilon$.*

*Sketch of proof.* The construction appearing in the proof of Theorem 5 with $L \equiv 0$ can be used to produce a radial neural network with $N(f, K, \epsilon)$ hidden layers with widths $n + m + 1$ that approximates $f$ on $K$. (Note that the approximation works only on $K$, as $f$ is not defined outside of $K$.) All values in the hidden layers are of the form $(x, 0, 0)$ or $(0, y, 1)$. We can therefore replace $(x, y, \theta) \in \mathbb{R}^{n+m+1}$ by $(x + y, \theta) \in \mathbb{R}^{\max(n,m)} \times \mathbb{R} \cong \mathbb{R}^{\max(n,m)+1}$ everywhere, without affecting any statements about the hidden layers. In particular, the transformation $T_i$ becomes

$$T_i : \mathbb{R}^{\max(n,m)+1} \to \mathbb{R}^{\max(n,m)+1}, \qquad (x, \theta) \mapsto \left(x - (1 - \theta)c_i - \theta\frac{f(c_i)}{s} , (1 - \theta)h_i\right).$$

With this change the final affine map $\Phi$ sends $(x, \theta)$ to $sx$. From the rest of the proof of Theorem 5 it follows that the feedforward function $F$ of the radial network satisfies $|F(x) - f(x)| < \epsilon$ for all $x \in K$. $\square$

## B.6 Additional result: bound of $\max(n, m)$

In this section, we prove a different version of the result of the previous section. Specifically, we reduce the bound on the widths to $\max(n, m)$ at the cost of using more layers. Again, we focus on functions defined on a compact domain without assumptions about their asymptotic behavior. Recall the notation $M(f, K, \epsilon)$ from Lemma 10 and Definition 11.

**Theorem 16.** *Let $f : K \to \mathbb{R}^m$ be a continuous function, where $K$ is a compact subset of $\mathbb{R}^n$ for $n \geq 2$. For any $\epsilon > 0$, there exists $F : \mathbb{R}^n \to \mathbb{R}^m$ such that:*

1. *$F$ is the feedforward function of a radial neural network with $2M(f, K, \epsilon/2)$ hidden layers whose widths are all $\max(n, m)$.*
2. *For any $x \in K$, we have $|F(x) - f(x)| < \epsilon$.*

*Proof.* We first consider the proof in the case $n = m$. Set $M = M(f, K, \epsilon)$. As in Lemma 10, fix $c_1, \ldots, c_M \in K$ and $r_1, \ldots, r_M \in (0, 1)$ such that, first, the union of the balls $B_{r_i}(c_i)$ covers $K$; second, for all $i$, we have $f(B_{r_i}(c_i)) \subseteq B_{\epsilon/2}(f(c_i))$; and third, $|c_i - c_j| \geq r_i$ for $i \neq j$. For $i = 1, \ldots, M$, set

$$T_i : \mathbb{R}^n \to \mathbb{R}^n, \qquad x \mapsto \frac{x - c_i}{r_i},$$

and recursively define $G_i : \mathbb{R}^n \to \mathbb{R}^n$ as $G_i = T_i^{-1} \circ \rho \circ T_i \circ G_{i-1}$, where $G_0 = \mathrm{id}_{\mathbb{R}^n}$ is the identity on $\mathbb{R}^n$ and $\rho : \mathbb{R}^n \to \mathbb{R}^n$ is Step-ReLU.

**Lemma 16.1.** *For $i = 0, 1, \ldots, N$, we have:*

$$G_i(x) = \begin{cases} x & \text{if } x \notin \bigcup_{j=1}^{i} B_{r_j}(c_j) \\ c_j & \text{where } j \leq i \text{ is the smallest index with } x \in B_{r_j}(c_j). \end{cases}$$

We omit the full proof of Lemma 16.1, as it is a standard induction argument similar to Proposition 3.2, relying on the following two facts. First, $|T_i(x)| < 1$ if and only if $x \in B_{r_i}(c_i)$. Second, by the choice of $c_i$, we have $|c_i - c_j| \geq r_i$ for all $i \neq j$. This implies that $|T_i(c_j)| \geq 1$ for $i \neq j$.

Next, perform the following loop over $i = 1, \ldots, M$:

- Set $P_{i-1} = \{c_1, \ldots, c_M\} \cup \{d_1, \ldots, d_{i-1}\}$

- Choose $d_i$ in $B_{\epsilon/2}(f(c_i))$ that is not colinear with any pair of points in $P_{i-1}$. This is where we use the hypothesis that $n \geq 2$.

- Let $s_i$ be the minimum distance between any point on the line through $c_i$ and $d_i$ and any point in $P_{i-1} \setminus \{c_i\}$.

- Let $U_i : \mathbb{R}^n \to \mathbb{R}^n$ be the following affine transformation:

$$U_i : \mathbb{R}^n \to \mathbb{R}^n, \qquad x \mapsto \frac{x - d_i}{s_i} + \left( \frac{1}{|c_i - d_i|} - \frac{1}{s_i} \right) \frac{\langle x - d_i, c_i - d_i \rangle}{|c_i - d_i|^2} (c_i - d_i)$$

- Define $H_i : \mathbb{R}^n \to \mathbb{R}^n$ recursively as $H_i = U_i^{-1} \circ \rho \circ U_i \circ H_{i-1}$, where $H_0 = \mathrm{id}_{\mathbb{R}^n}$.

We note that the transformation $U_i$ can also be written as $A_i(x - d_i)$ where $A_i$ is the linear map given by $A_i = \frac{1}{s_i} \mathrm{proj}_{\langle c_i - d_i \rangle^\perp} + \frac{1}{|c_i - d_i|} \mathrm{proj}_{\langle c_i - d_i \rangle}$, which involves the projections onto the line spanned by $c_i - d_i$ and onto the orthogonal complement of this line.

**Lemma 16.2.** *For $i, j = 1, \ldots, M$, we have:*

$$H_i(c_j) = \begin{cases} d_j & \text{if } j \leq i \\ c_j & \text{if } j > i \end{cases}$$

784 *Proof.* It is immediate that $U_i(d_i) = 0$ and $|U_i(c_i)| = 1/2$. It is also straightforward to show,
785 using the choice of $s_i$, that $|U_i(p)| \geq 1$ for all $p \in P_{i-1} \setminus \{c_i\}$. It follows that $U_i^{-1} \circ \rho \circ U_i$
786 sends $c_i$ to $d_i$ and fixes all other points in $P_{i-1}$. $\square$

787 **Lemma 16.3.** For $x \in K$, we have $H_M \circ G_M(x) = d_i$ where $i$ is the smallest index with
788 $x \in B_{r_i}(c_i)$

789 *Proof.* Let $x \in K$. By Lemma 16.1, we have that $G_M(x) = c_i$ where $i$ is the smallest index
790 with $x \in B_{r_i}(c_i)$. (We use the fact that the balls $\{B_{r_i}(c_i)\}$ cover $K$.) By Lemma 16.2, we have
791 that $H_M(c_i) = d_i$ for all $i$. The result follows. $\square$

792 Set $F = H_M \circ G_M$. We see that, for $x \in K$:

$$|F(x) - f(x)| = |d_i - f(x)| \leq |d_i - f(c_i)| + |f(c_i) - f(x)| < \epsilon/2 + \epsilon/2 = \epsilon$$

793 where $i$ is the smallest index with $x \in B_{r_i}(c_i)$. We show that $F$ is the feedforward function
794 of a radial neural network with $2M$ hidden layers, all of width equal to $n$. Indeed, take the
795 affine transformations and activations as follows:

796 • For $i = 1, \ldots, M$ the affine transformation from layer $i - 1$ to layer $i$ is given by
797    $x \mapsto T_i \circ T_{i-1}^{-1}(x)$, where $T_0 = \mathrm{id}_{\mathbb{R}^n}$.

798 • For $i = 1, \ldots, M$ the affine transformation from layer $M + i - 1$ to layer $M + i$ is
799    given by $x \mapsto U_i \circ U_{i-1}^{-1}(x)$, where $U_0 = T_N^{-1}$.

800 • The activation at each hidden layer is Step-ReLU on $\mathbb{R}^n$ that is $\rho(x) = x$ if $|x| \geq 1$
801    and 0 otherwise.

802 • Layer $2M + 1$ has the affine transformation $U_M^{-1}$.

803 It is immediate from definitions that the feedforward function of this network is $F$.

804 To conclude the proof, we discuss the cases where $n \neq m$. Suppose $n < m$ so that
805 $\max(n, m) = m$. Then we can regard $K$ as a compact subset of $\mathbb{R}^m$ and apply the above
806 constructions. Suppose $n > m$ so that $\max(n, m) = n$. Let $\mathrm{inc} : \mathbb{R}^m \hookrightarrow \mathbb{R}^n$. Apply the
807 above constructions to the function $\tilde{f} = \mathrm{inc} \circ f : K \to \mathbb{R}^n$. $\square$

# C Model compression proofs

809 The aim of this appendix is to give a proof of Theorem 6. In order to do so, we first (1)
810 provide background on a relevant version of the QR decomposition, and (2) establish basic
811 properties of radial rescaling activations.

## C.1 The QR decomposition

813 In this section, we recall the QR decomposition and note several relevant facts. For integers
814 $n$ and $m$, let $(\mathbb{R}^{n \times m})^{\mathrm{upper}}$ denote the vector space of upper triangular $n$ by $m$ matrices.

815 **Theorem 17** (QR Decomposition)**.** *The following map is surjective:*

$$O(n) \times (\mathbb{R}^{n \times m})^{\mathrm{upper}} \longrightarrow \mathbb{R}^{n \times m}$$
$$Q, R \mapsto Q \circ R$$

816 In other words, any matrix can be written as the product of an orthogonal matrix and an
817 upper-triangular matrix. When $m \leq n$, the last $n - m$ rows of any matrix in $(\mathbb{R}^{n \times m})^{\mathrm{upper}}$
818 are zero, and the top $m$ rows form an upper-triangular $m$ by $m$ matrix. These observations
819 lead to the following "complete" version of the QR decomposition, which coincides with
820 the above result when $m \geq n$:

**Corollary 18** (Complete QR Decomposition). *The following map is surjective:*

$$\mu : O(n) \times \left(\mathbb{R}^{k \times m}\right)^{\text{upper}} \longrightarrow \mathbb{R}^{n \times m}$$
$$Q \, , \, R \quad \mapsto \quad Q \circ \text{inc} \circ R$$

*where $k = \min(n,m)$ and $\text{inc} : \mathbb{R}^k \hookrightarrow \mathbb{R}^n$ is the standard inclusion into the first $k$ coordinates.*

We make some remarks:

1. There are several algorithms for computing the QR decomposition of a given matrix. One is Gram–Schmidt orthogonalization, and another is the method of Householder reflections. The latter has computational complexity $O(n^2 m)$ in the case of a $n \times m$ matrix with $n \geq m$. The package numpy includes a function `numpy.linalg.qr` that computes the QR decomposition of a matrix using Householder reflections.

2. In each iteration of the loop in Algorithm 1, the method QR-decomp with mode = 'complete' takes as input a matrix $A_i$ of size $n_i \times (n_{i-1}^{\text{red}} + 1)$, and produces an orthogonal matrix $Q_i \in O(n_i)$ and an upper-triangular matrix $R_i$ of size $\min(n_i, n_{i-1}^{\text{red}} + 1) \times (n_{i-1}^{\text{red}} + 1)$ such that $A_i = Q_i \circ \text{inc}_i \circ R_i$. Note that $n_i^{\text{red}} = \min(n_i, n_{i-1}^{\text{red}} + 1)$.

3. The QR decomposition is not unique in general, or, in other words, the map $\mu$ is not injective in general. For example, if $n > m$, each fiber of $\mu$ contains a copy of the orthogonal group $O(n - m)$.

4. The QR decomposition is unique (in a certain sense) for invertible square matrices. To be precise, let $B_n^+$ be the subset of of $(\mathbb{R}^{n \times n})^{\text{upper}}$ consisting of upper triangular $n$ by $n$ matrices with positive entries along the diagonal. Both $B_n^+$ and $O(n)$ are subgroups of the general linear group $\text{GL}_n(\mathbb{R})$, and the multiplication map $O(n) \times B_n^+ \to \text{GL}_n(\mathbb{R})$ is bijective. However, the QR decomposition is not unique for non-invertible square matrices.

## C.2 Radial rescaling functions

We now prove the following basic facts about radial rescaling functions:

**Lemma 19.** *Let $\rho = h^{(n)} : \mathbb{R}^n \to \mathbb{R}^n$ be a radial rescaling function on $\mathbb{R}^n$.*

1. *The function $\rho$ commutes with any orthogonal transformation of $\mathbb{R}^n$. That is, $\rho \circ Q = Q \circ \rho$ for any $Q \in O(n)$.*

2. *If $m \leq n$ and $\text{inc} : \mathbb{R}^m \hookrightarrow \mathbb{R}^n$ is the standard inclusion into the first $m$ coordinates, then: $h^{(n)} \circ \text{inc} = \text{inc} \circ h^{(m)}$.*

*Proof.* Suppose $Q \in O(n)$ is an orthogonal transformation of $\mathbb{R}^n$. Since $Q$ is norm-preserving, we have $|Qv| = |v|$ for any $v \in \mathbb{R}^n$. Since $Q$ is linear, we have $Q(\lambda v) = \lambda Q v$ for any $\lambda \in \mathbb{R}$ and $v \in \mathbb{R}^n$. Using the definition of $a = h^{(n)}$ we compute:

$$\rho(Qv) = \frac{h(|Qv|)}{|Qv|} Qv = \frac{h(|v|)}{|v|} Qv = Q\left(\frac{h(|v|)}{|v|} v\right) = Q(\rho(v)).$$

The first claim follows. The second claim is an elementary verification. $\square$

More generally, the restriction of the radial rescaling function $\rho$ to a linear subspace of $\mathbb{R}^n$ is a radial rescaling function on that subspace. Given a tuple radial rescaling functions $\rho = (\rho_i : \mathbb{R}^{n_i} \to \mathbb{R}^{n_i})_{i=1}^L$ suited to widths $\mathbf{n} = (n_i)_{i=1}^L$, we write $\rho^{\text{red}} = \left(\rho_i^{\text{red}} : \mathbb{R}^{n_i^{\text{red}}} \to \mathbb{R}^{n_i^{\text{red}}}\right)$

for the tuple of restrictions suited to the reduced widths $\mathbf{n}^{\text{red}}$, so that $\rho_i^{\text{red}} = \rho_i\big|_{\mathbb{R}^{n_i^{\text{red}}}}$.

 **C.3  Proof of Theorem 6**

860  Adopting notation from above and Section 5, we now restate and prove Theorem 6.

861  **Theorem 6.** *Let* $(\mathbf{W}, \mathbf{b}, \boldsymbol{\rho})$ *be a radial neural network with widths* $\mathbf{n}$*. Let* $\mathbf{W}^{\text{red}}$ *and* $\mathbf{b}^{\text{red}}$ *be the*
862  *weights and biases of the compressed network produced by Algorithm 1. The feedforward function*
863  *of the original network* $(\mathbf{W}, \mathbf{b}, \boldsymbol{\rho})$ *coincides with that of the compressed network* $(\mathbf{W}^{\text{red}}, \mathbf{b}^{\text{red}}, \boldsymbol{\rho}^{\text{red}})$*.*

864  *Proof.* Let $(\mathbf{W}^{\text{red}}, \mathbf{b}^{\text{red}}, \mathbf{Q}) = \texttt{QR-Compress}(\mathbf{W}, \mathbf{b})$ be the output of Algorithm 1, so that
865  $\mathbf{Q} \in O(\mathbf{n}^{\text{hid}})$ and $(\mathbf{W}^{\text{red}}, \mathbf{b}^{\text{red}}, \boldsymbol{\rho}^{\text{red}})$ is a neural network with widths $n^{\text{red}}$ and radial
866  rescaling activations $\rho_i^{\text{red}} = \rho_i\big|_{\mathbb{R}^{n_i^{\text{red}}}}$. Let $F = F_{(\mathbf{W}, \mathbf{b}, \boldsymbol{\rho})}$ denote the feedforward function
867  of the radial neural network with parameters $(\mathbf{W}, \mathbf{b})$ and activations $\boldsymbol{\rho}$. Similarly, let
868  $F^{\text{red}} = F_{(\mathbf{W}^{\text{red}}, \mathbf{b}^{\text{red}}, \boldsymbol{\rho}^{\text{red}})}$ denote the feedforward function of the radial neural network with
869  parameters $(\mathbf{W}^{\text{red}}, \mathbf{b}^{\text{red}})$ and activations $\boldsymbol{\rho}^{\text{red}}$. Additionally, we have the partial feedforward
870  functions $F_i$ and $F_i^{\text{red}}$. We show by induction that

$$F_i = Q_i \circ \text{inc}_i \circ F_i^{\text{red}}$$

871  for any $i = 0, 1, \ldots, N$. (Continuing conventions from Sections 5.1 and 5.2, we set $Q_0 =$
872  $\text{id}_{\mathbb{R}^{n_0}}$, $Q_L = \text{id}_{\mathbb{R}^{n_L}}$, and $\text{inc}_i : \mathbb{R}^{n_i^{\text{red}}} \to \mathbb{R}^{n_i}$ to be the inclusion map.) The base step $i = 0$
873  immediate. For the induction step, let $x \in \mathbb{R}^{n_0}$. Then:

$$\begin{aligned}
F_i(x) &= \rho_i \left( W_i \circ F_{i-1}(x) + b_i \right) \\
&= \rho_i \left( W_i \circ Q_{i-1} \circ \text{inc}_{i-1} \circ F_{i-1}^{\text{red}}(x) + b_i \right) \\
&= \rho_i \left( \begin{bmatrix} b_i & W_i \circ Q_{i-1} \circ \text{inc}_{i-1} \end{bmatrix} \begin{bmatrix} 1 \\ F_{i-1}^{\text{red}}(x) \end{bmatrix} \right) \\
&= \rho_i \left( Q_i \circ \text{inc}_i \circ \begin{bmatrix} b_i^{\text{red}} & W_i^{\text{red}} \end{bmatrix} \begin{bmatrix} 1 \\ F_{i-1}^{\text{red}}(x) \end{bmatrix} \right) \\
&= Q_i \circ \text{inc}_i \circ \rho_i\big|_{\mathbb{R}^{n_i^{\text{red}}}} \left( W_i^{\text{red}} \circ F_{i-1}^{\text{red}}(x) + b_i^{\text{red}} \right) \\
&= Q_i \circ \text{inc}_i \circ F_i^{\text{red}}
\end{aligned}$$

874  The first equality relies on the definition of the partial feedforward function $F_i$; the second
875  on the induction hypothesis; the fourth on an inspection of Algorithm 1, noting that
876  $R_i = \begin{bmatrix} b_i^{\text{red}} & W_i^{\text{red}} \end{bmatrix}$; the fifth on the results of Lemma 19, observing that $\rho_i \circ \text{inc}_i = \rho_i\big|_{\mathbb{R}^{n_i^{\text{red}}}} =$
877  $\text{inc}_i \circ \rho_i^{\text{red}}$; and the sixth on the definition of $F_i^{\text{red}}$. In the case $i = L$, we have:

$$F = F_L = Q_L \circ \text{inc}_L \circ F_L^{\text{red}} = F^{\text{red}}$$

878  since $Q_L = \text{inc}_L = \text{id}_{\mathbb{R}^{n_L}}$ and $F_L^{\text{red}} = F^{\text{red}}$. The theorem now follows.  □

879  The techniques of the above proof can be used to show that the action of the group $O(\mathbf{n}^{\text{hid}})$
880  of orthogonal change-of-basis symmetries on the parameter space $\text{Param}(\mathbf{n})$ leaves the
881  feedforward function unchanged. We do not use this result directly, but state is precisely it
882  nonetheless:

883  **Proposition 20.** *Let* $(\mathbf{W}, \mathbf{b}, \boldsymbol{\rho})$ *be a radial neural network with widths vector* $\mathbf{n}$*. Suppose* $\mathbf{g} \in$
884  $O(\mathbf{n}^{\text{hid}})$*. Then the original and transformed networks have the same feedforward function:*

$$F_{(\mathbf{g} \cdot \mathbf{W}, \, \mathbf{g} \cdot \mathbf{b}, \, \boldsymbol{\rho})} = F_{(\mathbf{W}, \, \mathbf{b}, \, \boldsymbol{\rho})}$$

885  In other words, fix parameters $(\mathbf{W}, \mathbf{b}) \in \text{Param}(\mathbf{n})$, radial rescaling activations $\boldsymbol{\rho}$, and $\mathbf{g} \in$
886  $O(\mathbf{n}^{\text{hid}})$. Then the radial neural network with parameters $(\mathbf{W}, \mathbf{b})$ has the same feedforward

function as the radial neural network with transformed parameters $(\mathbf{g} \cdot \mathbf{W}, \mathbf{g} \cdot \mathbf{b})$, where we take radial rescaling activations $\rho$ in both cases.

We remark that Proposition 20 is analogous to the "non-negative homogeneity" (or "positive scaling invariance") of the pointwise ReLU activation function[3]. In that setting, instead of considering the product of orthogonal groups $O(\mathbf{n}^{\mathrm{hid}})$, one considers the rescaling action of the following subgroup of $\prod_{i=1}^{L-1} \mathrm{GL}_{n_i}$:

$$G = \left\{ \mathbf{g} = (g_i) \in \prod_{i=1}^{L-1} \mathrm{GL}_{n_i} \mid \text{ each } g_i \text{ is diagonal with positive diagonal entries} \right\}$$

Note that $G$ is isomorphic to the product $\prod_{i=1}^{L-1} \mathbb{R}_{>0}^{n_i}$, and the action on Param($\mathbf{n}$) is given by the same formulas as those appearing near the end of Section 5.1. The feedforward function of a MLP with pointwise ReLU activations is invariant for the action of $G$ on Param($\mathbf{n}$).

# D   Projected gradient descent proofs

In this section, we give a proof of Theorem 8, which relates projected gradient descent for a representation with dimension $\mathbf{n}$ to (usual) gradient descent for the corresponding reduced representation with dimension vector $\mathbf{n}^{\mathrm{red}}$. This proof requires some set up and background resutls.

## D.1   Gradient descent and orthogonal symmetries

We first prove a result that gradient descent commutes with invariant orthogonal transformations. This section is general and departs from the specific case of radial neural networks.

### D.1.1   Setting

Let $\mathcal{L} : V = \mathbb{R}^p \to \mathbb{R}$ be a smooth function. Semantically, $V$ is a the parameter space of a neural network and $\mathcal{L}$ the loss function with respect to a batch of training data. The differential $d\mathcal{L}_v$ of $\mathcal{L}$ at $v \in V$ is row vector, while the gradient $\nabla_v \mathcal{L}$ of $\mathcal{L}$ at $v$ is a column vector[4]:

$$d\mathcal{L}_v = \begin{bmatrix} \left.\frac{\partial \mathcal{L}}{\partial x_1}\right|_v & \cdots & \left.\frac{\partial \mathcal{L}}{\partial x_p}\right|_v \end{bmatrix} \qquad \nabla_v \mathcal{L} = \begin{bmatrix} \left.\frac{\partial \mathcal{L}}{\partial x_1}\right|_v \\ \vdots \\ \left.\frac{\partial \mathcal{L}}{\partial x_p}\right|_v \end{bmatrix}$$

Hence $\nabla_v \mathcal{L}$ is the transpose of $d\mathcal{L}_v$, that is: $\nabla_v \mathcal{L} = (d\mathcal{L}_v)^T$. A step of gradient descent with respect to $\mathcal{L}$ at learning rate $\eta > 0$ is defined as:

$$\gamma = \gamma_\eta : V \longrightarrow V$$
$$v \longmapsto v - \eta \nabla_v \mathcal{L}$$

---

[3]See Armenta and Jodoin, *The Representation Theory of Neural Networks*, arXiv:2007.12213; Dinh, Pascanu, Bengio, and Bengio, *Sharp Minima Can Generalize For Deep Nets*, ICML 2017; Meng, Zheng, Zhang, Chen, Ye, Ma, Yu, and Liu, *G-SGD: Optimizing ReLU Neural Networks in its Positively Scale-Invariant Space*, 2019; and Neyshabur, Salakhutdinov, and Srebro. *Path-SGD: path-normalized optimization in deep neural networks*, NIPS'15.

[4]Following usual conventions, we regard column vectors as elements of $V$ and row vectors as elements of the dual vector space $V^*$. The differential $d\mathcal{L}_v$ of $\mathcal{L}$ at $v \in V$ is also known as the Jacobian of $\mathcal{L}$ at $v \in V$.

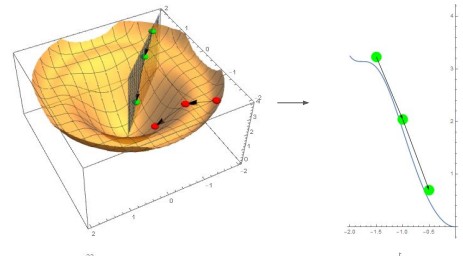

Figure 5: Illustration of Lemma 22. If the loss is invariant with respect to an orthogonal transformation $Q$ of the parameter space, then optimization of the network by gradient descent is also invariant with respect to $Q$. (Note: in this example, projected and usual gradient descent match; this is not the case in higher dimensions, as explained in D.6.)

We drop $\eta$ from the notation when it is clear from context. For any $k \geq 0$, we denote by $\gamma^k$ the $k$-fold composition of the gradient descent map $\gamma$:

$$\gamma^k = \overbrace{\gamma \circ \gamma \circ \cdots \circ \gamma}^{k}$$

### D.1.2 Invariant group action

Now suppose $\rho : G \to \mathrm{GL}(V)$ is an action of a Lie group $G$ on $V$ such that $\mathcal{L}$ is $G$-invariant, i.e.:

$$\mathcal{L}(\rho(g)(v)) = \mathcal{L}(v)$$

for all $g \in G$ and $v \in V$. We write simply $g \cdot v$ for $\rho(g)(v)$, and $g$ for $\rho(g)$.

**Lemma 21.** *For any $v \in V$ and $g \in G$, we have:*

$$\nabla_v \mathcal{L} = g^T \cdot (\nabla_{g \cdot v} \mathcal{L})$$

*Proof.* The proof is a computation:

$$\nabla_v \mathcal{L} = (d_v \mathcal{L})^T = (d(\mathcal{L} \circ g)_v)^T = (d\mathcal{L}_{g \cdot v} \circ dg_v)^T = (d\mathcal{L}_{g \cdot v} \circ g)^T = g^T \cdot (d\mathcal{L}_{g \cdot v})^T$$
$$= g^T \cdot (\nabla \mathcal{L}_{g \cdot v})$$

The second equality relies on the hypothesis that $\mathcal{L} \circ g = \mathcal{L}$, the third on the chain rule, and the fourth on the fact that $dg_v = g$ since $g$ is a linear map. $\qquad\square$

One can perform the computation of the proof in coordinates, for $i = 1, \ldots, p$:

$$(\nabla_v \mathcal{L})_i = (d\mathcal{L}_v)^i = \left.\frac{\partial \mathcal{L}}{\partial x_i}\right|_v = \left.\frac{\partial (\mathcal{L} \circ g)}{\partial x_i}\right|_v = \left.\frac{\partial \mathcal{L}}{\partial x_j}\right|_{gv} \left.\frac{\partial g_j}{\partial x_i}\right|_v$$
$$= (\nabla_{gv} \mathcal{L})_j \, g_j^i = (g^T)_i^j \, (\nabla_{gv} \mathcal{L})_j = \left(g^T \cdot \nabla_{gv} \mathcal{L}\right)_i$$

### D.1.3 Orthogonal case

Furthermore, suppose the action of $G$ is by orthogonal transformations, so that $\rho(g)^T = \rho(g)^{-1}$ for all $g \in G$. Then Lemma 21 implies that

$$\nabla_{g \cdot v} \mathcal{L} = g \cdot \nabla_v \mathcal{L} \qquad\qquad (\text{D.1})$$

for any $v \in V$ and $g \in G$. The proof of the following lemma is immediate from Equation D.1, together with the definition of $\gamma$. See Figure 5 for an illustration.

**Lemma 22.** *Suppose the action of $G$ on $V$ is by orthogonal transformations, and that $\mathcal{L}$ is $G$-invariant. Then the action of $G$ commutes with gradient descent (for any learning rate). That is,*

$$\gamma^k(g \cdot v) = g \cdot \gamma^k(v)$$

*for any $v \in V$, $g \in G$, and $k \geq 0$.*

## D.2 Gradient descent notation and set-up

We now turn our attention back to radial neural networks. In this section, we recall notation from above, and introduce new notation that will be relevant for the formulation and proof of Theorem 8.

### D.2.1 Merging widths and biases

Let $\mathbf{n} = (n_0, n_1, n_2, \ldots, n_{L-1}, n_L)$ be the widths vector of an MLP. Recall the definition of $\mathsf{Param}(\mathbf{n})$ as the parameter space of all possible choices of trainable parameters:

$$\mathsf{Param}(\mathbf{n}) = \left( \mathbb{R}^{n_1 \times n_0} \times \mathbb{R}^{n_2 \times n_1} \times \cdots \times \mathbb{R}^{n_L \times n_{L-1}} \right) \times \left( \mathbb{R}^{n_1} \times \mathbb{R}^{n_2} \times \cdots \times \mathbb{R}^{n_L} \right)$$

We have been denoting an element therein as a pair of tuples $(\mathbf{W}, \mathbf{b})$ where $\mathbf{W} = (W_i \in \mathbb{R}^{n_i \times n_{i-1}})_{i=1}^L$ are the weights and $\mathbf{b} = (b_i \in \mathbb{R}^{n_i})_{i=1}^L$ are the biases. However, in this appendix we adopt different notation. Observe that, placing each bias vector as a extra column on the left of the weight matrix, we obtain matrices:

$$A_i = [b_i \; W_i] \;\in\; \mathbb{R}^{n_i \times (1 + n_{i-1})}.$$

Thus, there is an isomorphism:

$$\mathsf{Param}(\mathbf{n}) \simeq \bigoplus_{i=1}^L \mathbb{R}^{n_i \times (n_{i-1}+1)} = \mathbb{R}^{n_1 \times (n_0+1)} \times \mathbb{R}^{n_2 \times (n_1+1)} \times \cdots \times \mathbb{R}^{n_L \times (n_{L-1}+1)}$$

In this appendix, we regard an element of $\mathsf{Param}(\mathbf{n})$ as a tuple of 'merged' matrices $\mathbf{A} = (A_i \in \mathbb{R}^{n_i \times (1+n_{i-1})})_{i=1}^L$. We now define convenient maps to translate between the merged notation and the split notation. For each $i$, define the extension-by-one map from $\mathbb{R}^{n_i}$ to $\mathbb{R} \times \mathbb{R}^{n_i} \simeq \mathbb{R}^{n_i+1}$ as follows:

$$\mathrm{ext}_i : \mathbb{R}^{n_i} \to \mathbb{R}^{n_i+1} \qquad v = (v_1, v_2, \ldots, v_{n_i}) \mapsto (1, v_1, v_2, \ldots, v_{n_i}) \tag{D.2}$$

Observe that, for any $i$ and $x \in \mathbb{R}^{n_{i-1}}$, we have

$$A_i \circ \mathrm{ext}_{i-1}(x) = W_i x + b_i.$$

Consequently, the $i$-th partial feedforward function can be defined recursively as:

$$F_i = \rho_i \circ A_i \circ \mathrm{ext}_{i-1} \circ F_{i-1} \tag{D.3}$$

where $\rho_i : \mathbb{R}^{n_i} \to \mathbb{R}^{n_i}$ is the activation[5] at the $i$-th layer, and $F_0$ is the identity on $\mathbb{R}^{n_0}$.

### D.2.2 Orthogonal change-of-basis action

To describe the orthogonal change-of-basis symmetries of the parameter space in the merged notation, recall the following product of orthogonal groups, with sizes corresponding to the widths of the hidden layers:

$$O(\mathbf{n}^{\mathrm{hid}}) = \mathcal{O}(n_1) \times O(n_2) \times \cdots \times O(n_{L-1})$$

In the merged notation, the element $\mathbf{Q} = (Q_i)_{i=1}^{L-1} \in O(\mathbf{n}^{\mathrm{hid}})$ transforms $\mathbf{A} \in \mathsf{Param}(\mathbf{n})$ as:

$$\mathbf{A} \quad \mapsto \quad \mathbf{Q} \cdot \mathbf{A} := \left( Q_i \circ A_i \circ \begin{bmatrix} 1 & 0 \\ 0 & Q_{i-1}^{-1} \end{bmatrix} \right)_{i=1}^L \tag{D.4}$$

where $Q_0 = \mathrm{id}_{n_0}$ and $Q_L = \mathrm{id}_{n_L}$.

---

[5]In this general formulation, $\rho_i$ can be any piece-wise differentiable function; for most of the rest of the paper we will be interested in the case where $\rho_i$ is a radial rescaling function.

### D.2.3 Model compression algorithm

We now restate Algorithm 1 in the merged notation. We emphasize that Algorithms 1 and 2 are mathematically equivalent; the later simply uses more compact notation.

---

**Algorithm 2:** QR Model Compression (`QR-compress`)

---

**input**    : $\mathbf{A} \in \mathsf{Param}(\mathbf{n})$
**output**   : $\mathbf{Q} \in O(\mathbf{n}^{\text{hidden}})$ and $\mathbf{V} \in \mathsf{Param}(\mathbf{n}^{\text{red}})$

$\mathbf{Q}, \mathbf{V} \leftarrow [\,], [\,]$                               // initialize output matrix lists
$M_1 \leftarrow A_1$
**for** $i \leftarrow 1$ **to** $L - 1$ **do**                      // iterate through layers
    $Q_i, R_i \leftarrow$ `QR-decomp`$(M_i,$ *mode = 'complete'*$)$         // $M_i = Q_i \circ \text{inc}_i \circ R_i$
    Append $Q_i$ to $\mathbf{Q}$
    Append $R_i$ to $\mathbf{V}$                   // reduced merged weights for layer $i$

    Set $M_{i+1} \leftarrow A_{i+1} \circ \begin{bmatrix} 1 & 0 \\ 0 & Q_i \circ \text{inc}_i \end{bmatrix}$         // transform next layer
**end**
Append $M_L$ to $\mathbf{V}$

**return** $\mathbf{Q}, \mathbf{V}$

---

We explain the notation. As noted in Appendix B.1, the symbol '$\circ$' denotes composition of maps, or matrix multiplication in the case of linear maps. The standard inclusion $\text{inc}_i : \mathbb{R}^{n_i^{\text{red}}} \hookrightarrow \mathbb{R}^{n_i}$ maps into the first $n_i^{\text{red}}$ coordinates. As a matrix, $\text{Inc}_i \in \mathbb{R}^{n_i \times n_i^{\text{red}}}$ has ones along the main diagonal and zeros elsewhere. The method `QR-decomp` with mode = 'complete' computes the complete QR decomposition of the $n_i \times (1 + n_{i-1}^{\text{red}})$ matrix $M_i$ as $Q_i \circ \text{inc}_i \circ R_i$ where $Q_i \in O(n_i)$ and $R_i$ is upper-triangular of size $n_i^{\text{red}} \times (1 + n_{i-1}^{\text{red}})$. The definition of $n_i^{\text{red}}$ implies that either $n_i^{\text{red}} = n_{i-1}^{\text{red}} + 1$ or $n_i^{\text{red}} = n_i$. The matrix $R_i$ is of size $n_i^{\text{red}} \times n_i^{\text{red}}$ in the former case and of size $n_i \times (1 + n_{i-1}^{\text{red}})$ in the latter case.

### D.2.4 Gradient descent definitions

As in Section 6, we fix:

- a widths vector $\mathbf{n} = (n_0, n_1, \ldots, n_L)$.
- a tuple $\boldsymbol{\rho} = (\rho_1, \ldots, \rho_L)$ of radial rescaling activations, where $\rho_i : \mathbb{R}^{n_i} \to \mathbb{R}^{n_i}$ for $i = 1, \ldots, L$.
- a batch of training data $\{(x_j, y_j)\} \subseteq \mathbb{R}^{n_0} \times \mathbb{R}^{n_L} = \mathbb{R}^{n_0^{\text{red}}} \times \mathbb{R}^{n_L^{\text{red}}}$.
- a cost function $\mathcal{C} : \mathbb{R}^{n_L} \times \mathbb{R}^{n_L} \to \mathbb{R}$

As a result, we have a loss function on $\mathsf{Param}(\mathbf{n})$:

$$\mathcal{L} : \mathsf{Param}(\mathbf{n}) \to \mathbb{R} \qquad \mathcal{L}(\mathbf{A}) = \sum \mathcal{C}(F_{(\mathbf{A}, \boldsymbol{\rho})}(x_j), y_j)$$

where $F_{(\mathbf{A}, \boldsymbol{\rho})}$ is the feedforward of the radial neural network with (merged) parameters $\mathbf{A}$ and activations $\boldsymbol{\rho}$. We emphasize that the loss function $\mathcal{L}$ depends on the batch of training data chosen above; however, for clarity, we omit extra notation indicating this dependency since the batch of training data is fixed throughout this discussion. Similarly, we have:

- the reduced widths vector $\mathbf{n}^{\text{red}} = (n_0^{\text{red}}, n_1^{\text{red}}, \ldots, n_L^{\text{red}})$.
- the restrictions $\boldsymbol{\rho}^{\text{red}} = (\rho_1^{\text{red}}, \ldots, \rho_L^{\text{red}})$, where $\rho_i^{\text{red}} : \mathbb{R}^{n_i^{\text{red}}} \to \mathbb{R}^{n_i^{\text{red}}}$ for $i = 1, \ldots, L$.

Using the fact that $n_0^{\text{red}} = n_0$ and $n_L^{\text{red}} = n_L$, there is a loss function on $\mathsf{Param}(\mathbf{n}^{\text{red}})$:

$$\mathcal{L}_{\text{red}} : \mathsf{Param}(\mathbf{n}^{\text{red}}) \to \mathbb{R} \qquad \mathcal{L}_{\text{red}}(\mathbf{B}) = \sum \mathcal{C}(F_{(\mathbf{B}, \boldsymbol{\rho}^{\text{red}})}(x_j), y_j)$$

28

where $F_{(\mathbf{B}, \rho^{\mathrm{red}})}$ is the feedforward of the radial neural network with parameters $\mathbf{B} \in$ Param$(\mathbf{n}^{\mathrm{red}})$ and activations $\rho^{\mathrm{red}}$. (Again, technically speaking, the loss function $\mathcal{L}_{\mathrm{red}}$ depends on the batch of training data fixed above.) For any learning rate $\eta > 0$, we obtain a gradient descent maps:

$$\gamma : \mathsf{Param}(\mathbf{n}) \to \mathsf{Param}(\mathbf{n}) \qquad\qquad \gamma_{\mathrm{red}} : \mathsf{Param}(\mathbf{n}^{\mathrm{red}}) \to \mathsf{Param}(\mathbf{n}^{\mathrm{red}})$$
$$\mathbf{A} \mapsto \mathbf{A} - \eta \nabla_{\mathbf{A}} \mathcal{L} \qquad\qquad\qquad \mathbf{B} \mapsto \mathbf{B} - \eta \nabla_{\mathbf{B}} \mathcal{L}_{\mathrm{red}}$$

## D.3 The interpolating space

In this section, we introduce a subspace $\mathsf{Param}^{\mathrm{int}}(\mathbf{n})$ of $\mathsf{Param}(\mathbf{n})$, that, as we will later see, interpolates between $\mathsf{Param}(\mathbf{n})$ and $\mathsf{Param}(\mathbf{n}^{\mathrm{red}})$.

Let $\mathsf{Param}^{\mathrm{int}}(\mathbf{n})$ denote the subspace of $\mathsf{Param}(\mathbf{n})$ consisting of those $\mathbf{T} = (T_1, \dots, T_L) \in$ $\mathsf{Param}(\mathbf{n})$ for which the bottom left $(n_i - n_i^{\mathrm{red}}) \times (1 + n_{i-1}^{\mathrm{red}})$ block of $T_i$ is zero for each $i$. Schematically:

$$T_i = \begin{bmatrix} * & * \\ 0 & * \end{bmatrix}$$

where the rows are divided as $n_i^{\mathrm{red}}$ on top and $n_i - n_i^{\mathrm{red}}$ on the bottom, while the columns are divided as $(1 + n_{i-1}^{\mathrm{red}})$ on the left and $n_{i-1} - n_{i-1}^{\mathrm{red}}$ on the right. Let

$$\iota_1 : \mathsf{Param}^{\mathrm{int}}(\mathbf{n}) \hookrightarrow \mathsf{Param}(\mathbf{n})$$

be the inclusion. The following proposition follows from an elementary analysis of the workings of Algorithm 2 (or, equivalently, Algorithm 1).

**Proposition 23.** *Let* $\mathbf{A} \in \mathsf{Param}(\mathbf{n})$ *and let* $\mathbf{Q} \in O(\mathbf{n}^{\mathrm{hid}})$ *be the tuple of orthogonal matrices produced by Algorithm 2. Then* $\mathbf{Q}^{-1} \cdot \mathbf{A}$ *belongs to* $\mathsf{Param}^{\mathrm{int}}(\mathbf{n})$.

Define a map

$$q_1 : \mathsf{Param}(\mathbf{n}) \to \mathsf{Param}^{\mathrm{int}}(\mathbf{n})$$

by taking $\mathbf{A} \in \mathsf{Param}(\mathbf{n})$ and zeroing out the bottom left $(n_i - n_i^{\mathrm{red}}) \times (1 + n_{i-1}^{\mathrm{red}})$ block of $A_i$ for each $i$. Schematically:

$$\mathbf{A} = \left( A_i = \begin{bmatrix} * & * \\ * & * \end{bmatrix} \right)_{i=1}^{L} \mapsto q_1(\mathbf{A}) = \left( \begin{bmatrix} * & * \\ 0 & * \end{bmatrix} \right)_{i=1}^{L}$$

It is straightforward to check that $q_1$ is a well-defined, surjective linear map. The transpose of $q_1$ is the inclusion $\iota_1$. We summarize the situation in the following diagram:

$$\mathsf{Param}^{\mathrm{int}}(\mathbf{n}) \underset{q_1}{\overset{\iota_1}{\rightleftarrows}} \mathsf{Param}(\mathbf{n}) \tag{D.5}$$

We observe that the composition $q_1 \circ \iota$ is the identity on $\mathsf{Param}^{\mathrm{int}}(\mathbf{n})$.

## D.4 Projected gradient descent and model compression

Recall from Section 6 that the *projected gradient descent* map on $\mathsf{Param}(\mathbf{n})$ is given by:

$$\gamma_{\mathrm{proj}} : \mathsf{Param}(\mathbf{n}) \to \mathsf{Param}(\mathbf{n}), \qquad \mathbf{A} \mapsto \mathrm{Proj}\left( \mathbf{A} - \eta \nabla_{\mathbf{A}} \mathcal{L} \right)$$

where $\mathbf{A} = (\mathbf{W}, \mathbf{b})$ are the merged parameters (Appendix D.2), and, in the notation of the previous section, the map Proj is $\iota_1 \circ q_1$. To reiterate, while all entries of each weight matrix and each bias vector contribute to the computation of the gradient $\nabla_{\mathbf{A}} \mathcal{L} = \nabla_{(\mathbf{W}, \mathbf{b})} \mathcal{L}$, only those not in the bottom left submatrix get updated under the projected gradient descent map $\gamma_{\mathrm{proj}}$.

Let $\mathbf{V}, \mathbf{Q} = \mathtt{QR\text{-}Compress}(\mathbf{A})$ be the outputs of Algorithm 2 (which is equivalent to Algorithm 1), so that $\mathbf{V} = (\mathbf{W}^{\mathrm{red}}, \mathbf{b}^{\mathrm{red}}) \in \mathrm{Param}(\mathbf{n}^{\mathrm{red}})$ are the parameters of the compressed model corresponding to the full model with merged parameters $\mathbf{A} = (\mathbf{W}, \mathbf{b})$, and $\mathbf{Q} \in O(\mathbf{n}^{\mathrm{hid}})$ is an orthogonal change-of-basis symmetry of the parameter space. Moreover, set $\mathbf{T} = \mathbf{Q}^{-1} \cdot \mathbf{A} \in \mathrm{Param}^{\mathrm{int}}(\mathbf{n})$, where we use the change-of-basis action from Appendix D.2 and Proposition 23. We have the following rephrasing of Theorem 8.

**Theorem 24** (Theorem 8)**.** *Let $\mathbf{A} \in \mathrm{Param}(\mathbf{n})$, and let $\mathbf{V}, \mathbf{Q}, \mathbf{T}$ be as above. For any $k \geq 0$:*

    *1.* $\gamma^k(\mathbf{A}) = \mathbf{Q} \cdot \gamma^k(\mathbf{T})$

    *2.* $\gamma^k_{\mathrm{proj}}(\mathbf{T}) = \gamma^k_{\mathrm{red}}(\mathbf{V}) + \mathbf{T} - \mathbf{V}$.

More precisely, the second equality is $\gamma^k_{\mathrm{proj}}(\mathbf{T}) = \iota(\gamma^k_{\mathrm{red}}(\mathbf{V})) + \mathbf{T} - \iota(\mathbf{V})$ where $\iota :$ $\mathrm{Param}(\mathbf{n}^{\mathrm{red}}) \hookrightarrow \mathrm{Param}(\mathbf{n})$ is the inclusion into the top left corner in each coordinate. Also, in the statement of Theorem 8, we have $\mathbf{U} = \mathbf{T} - \mathbf{V}$.

We summarize this result in the following diagram. The left horizontal maps indicate the addition of $\mathbf{U} = \mathbf{T} - \mathbf{V}$, the right horizontal arrows indicate the action of $\mathbf{Q}$, and the vertical maps are various versions of gradient descent. The shaded regions indicate the (smallest) vector space to which the various representations naturally belong.

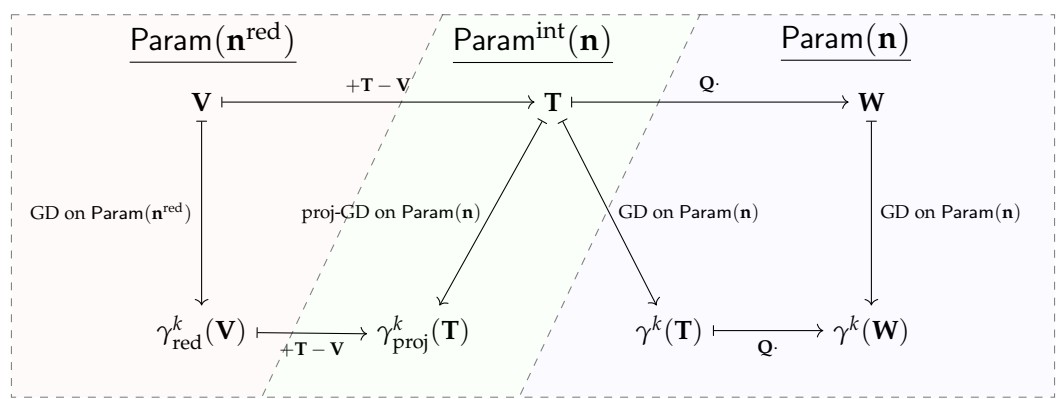

## D.5 Proof of Theorem 8

We begin by explaining the sense in which $\mathrm{Param}^{\mathrm{int}}(\mathbf{n})$ interpolates between $\mathrm{Param}(\mathbf{n})$ and $\mathrm{Param}(\mathbf{n}^{\mathrm{red}})$. One extends Diagram D.5 as follows:

$$\mathrm{Param}(\mathbf{n}^{\mathrm{red}}) \underset{q_2}{\overset{\iota_2}{\rightleftarrows}} \mathrm{Param}^{\mathrm{int}}(\mathbf{n}) \underset{q_1}{\overset{\iota_1}{\rightleftarrows}} \mathrm{Param}(\mathbf{n})$$

- The map
$$\iota_2 : \mathrm{Param}(\mathbf{n}^{\mathrm{red}}) \hookrightarrow \mathrm{Param}^{\mathrm{int}}(\mathbf{n})$$
takes $\mathbf{B} = (B_i) \in \mathrm{Param}(\mathbf{n}^{\mathrm{red}})$ and pad each matrix with $n_i - n_i^{\mathrm{red}}$ rows of zeros on the bottom and $n_{i-1} - n_{i-1}^{\mathrm{red}}$ columns of zeros on the right:
$$\mathbf{B} = (B_i)_{i=1}^L \mapsto \iota_2(\mathbf{B}) = \left( \begin{bmatrix} B_i & 0 \\ 0 & 0 \end{bmatrix} \right)_{i=1}^L$$
It is straightforward to check that $\iota_2$ is a well-defined injective linear map.

- The map
$$q_2 : \mathrm{Param}^{\mathrm{int}}(\mathbf{n}) \twoheadrightarrow \mathrm{Param}(\mathbf{n}^{\mathrm{red}})$$

extracts from $\mathbf{T}$ the top left $n_i^{\mathrm{red}} \times (1 + n_{i-1}^{\mathrm{red}})$ matrix:

$$\mathbf{T} = \left( T_i = \begin{bmatrix} T_i^{(1)} & T_i^{(2)} \\ 0 & T_i^{(4)} \end{bmatrix} \right)_{i=1}^{L} \mapsto q_2(\mathbf{T}) = \left( T_i^{(1)} \right)_{i=1}^{L}$$

It is straightforward to check that $q_2$ is a surjective linear map. The transpose of $q_2$ is the inclusion $\iota_2$.

**Lemma 25.** *We have the following:*

1. *The inclusion $\iota : \mathrm{Param}(\mathbf{n}^{\mathrm{red}}) \hookrightarrow \mathrm{Param}(\mathbf{n})$ coincides with the composition $\iota_1 \circ \iota_2$, and commutes with the loss functions:*

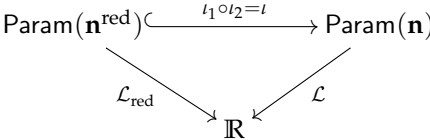

2. *The following diagram commutes:*

$$
\begin{array}{ccc}
\mathrm{Param}^{\mathrm{int}}(\mathbf{n}) & \xrightarrow{\ q_2\ } & \mathrm{Param}(\mathbf{n}^{\mathrm{red}}) \\
\Big\uparrow{\scriptstyle \iota_1} & & \Big\downarrow{\scriptstyle \mathcal{L}_{\mathrm{red}}} \\
\mathrm{Param}(\mathbf{n}) & \xrightarrow{\ \mathcal{L}\ } & \mathbb{R}
\end{array}
$$

3. *For any $\mathbf{T} \in \mathrm{Param}^{\mathrm{int}}(\mathbf{n})$, we have: $q_1 \left( \nabla_{\iota_1(\mathbf{T})} \mathcal{L} \right) = \iota_2 \left( \nabla_{q_2(\mathbf{T})} \mathcal{L}_{\mathrm{red}} \right)$.*

*Proof.* We have the following standard inclusions into the first coordinates and projections onto the first coordinates, for $i = 0, 1, \ldots, L$:

$$\mathrm{inc}_i = \mathrm{inc}_{n_i^{\mathrm{red}}, n_i} : \mathbb{R}^{n_i^{\mathrm{red}}} \hookrightarrow \mathbb{R}^{n_i}, \qquad \widetilde{\mathrm{inc}}_i = \mathrm{inc}_{1+n_i^{\mathrm{red}}, 1+n_i} : \mathbb{R}^{1+n_i^{\mathrm{red}}} \hookrightarrow \mathbb{R}^{1+n_i},$$

$$\pi_i : \mathbb{R}^{n_i} \twoheadrightarrow \mathbb{R}^{n_i^{\mathrm{red}}}, \qquad \widetilde{\pi}_i : \mathbb{R}^{1+n_i} \twoheadrightarrow \mathbb{R}^{1+n_i^{\mathrm{red}}}.$$

Observe that $\mathrm{Param}^{\mathrm{int}}(\mathbf{n})$ is the subspace of $\mathrm{Param}(\mathbf{n})$ consisting of those $\mathbf{T} = (T_1, \ldots, T_L) \in \mathrm{Param}(\mathbf{n})$ such that:

$$(\mathrm{id}_{n_i} - \mathrm{inc}_i \circ \pi_i) \circ T_i \circ \widetilde{\mathrm{inc}}_{i-1} \circ \widetilde{\pi}_{i-1} = 0$$

for $i = 1, \ldots, L$.

By the definition of radial rescaling functions, for each $i = 1, \ldots, L$, there is a piece-wise differentiable function $h_i : \mathbb{R} \to \mathbb{R}$ such that $\rho_i = h_i^{(n_i)}$. Note that $\rho_i^{\mathrm{red}} = h_i^{(n_i^{\mathrm{red}})}$, and $h^{(n_i)} \circ \mathrm{inc}_i = \mathrm{inc}_i \circ h^{(n_i^{\mathrm{red}})}$.

The identity $\iota = \iota_1 \circ \iota_2$ follows directly from definitions. To prove the commutativity of the first diagram, it is enough to show that, for any $\mathbf{X}$ in $\mathrm{Param}(\mathbf{n}^{\mathrm{red}})$, the feedforward functions of $\mathbf{X}$ and $\iota(\mathbf{X})$ coincide. This follows easily from the fact that, for $i = 1, \ldots, L$, we have:

$$\pi_i \circ h^{(n_i)} \circ \mathrm{inc}_i = \pi_i \circ \mathrm{inc}_i \circ h^{(n_i^{\mathrm{red}})} = h^{(n_i^{\mathrm{red}})}.$$

For the second claim, let $\mathbf{T} \in \mathrm{Param}^{\mathrm{int}}(\mathbf{n})$. It suffices to show that $\iota_1(\mathbf{T})$ and $q_2(\mathbf{T})$ have the same feedforward function. Recall the $\mathrm{ext}_i$ maps and the formulation of the feedforward function in the merged notation given in Equation D.3. Using this set-up, the key computation is:

$$\mathrm{inc}_i \circ h^{(n_i^{\mathrm{red}})} \circ \pi_i \circ T_i \circ \mathrm{ext}_{n_{i-1}} \circ \mathrm{inc}_{i-1} = h^{(n_i)} \circ \mathrm{inc}_i \circ \pi_i \circ T_i \circ \widetilde{\mathrm{inc}}_{i-1} \circ \mathrm{ext}_{n_{i-1}}$$

$$= h^{(n_i)} \circ T_i \circ \widetilde{\mathrm{inc}}_{i-1} \circ \mathrm{ext}_{n_{i-1}}$$

$$= h^{(n_i)} \circ T_i \circ \mathrm{ext}_{n_{i-1}} \circ \mathrm{inc}_{i-1}$$

which uses the fact that $(\mathrm{id}_{n_i} - \mathrm{inc}_i \circ \pi_i) \circ T_i \circ \widetilde{\mathrm{inc}}_{i-1} = 0$, or, equivalently, $\mathrm{inc}_i \circ \pi_i \circ T_i \circ \widetilde{\mathrm{inc}}_{i-1} = T_i \circ \widetilde{\mathrm{inc}}_{i-1}$, as well as the fact that $\mathrm{ext}_i \circ \mathrm{inc}_i = \widetilde{\mathrm{inc}}_i \circ \mathrm{ext}_i$. Applying this relation successively starting with the second-to-last layer ($i = L - 1$) and ending in the first ($i = 1$), one obtains the result. For the last claim, one computes $\nabla_{\mathbf{T}}(\mathcal{L} \circ \iota_1)$ in two different ways. The first way is:

$$\nabla_{\mathbf{T}}(\mathcal{L} \circ \iota_1) = (d(\mathcal{L}_{\mathbf{T}} \circ \iota_1))^T = \left(d\mathcal{L}_{\iota_1(\mathbf{T})} \circ d_{\mathbf{T}}\iota_1\right)^T = \left(d\mathcal{L}_{\iota_1(\mathbf{T})} \circ \iota_1\right)^T$$
$$= \iota_1^T\left(d\mathcal{L}_{\iota_1(\mathbf{T})}^T\right) = q_1\left(\nabla_{\iota_1(\mathbf{T})}\mathcal{L}\right)$$

where we use the fact that $\iota_1$ is a linear map whose transpose is $q_1$. The second way uses the commutative diagram of the second part of the Lemma:

$$\nabla_{\mathbf{T}}(\mathcal{L} \circ \iota_1) = \nabla_{\mathbf{T}}\left(\mathcal{L}_{\mathrm{red}} \circ q_2\right) = \left(d\left(\mathcal{L}_{\mathrm{red}}\right)_{\mathbf{T}} \circ q_2\right)^T = \left(d\left(\mathcal{L}_{\mathrm{red}}\right)_{q_2(\mathbf{T})} \circ d\left(q_2\right)_{\mathbf{Z}}\right)^T$$
$$= \left(d\left(\mathcal{L}_{\mathrm{red}}\right)_{q_2(\mathbf{T})} \circ q_2\right)^T = q_2^T\left(d\left(\mathcal{L}_{\mathrm{red}}\right)_{q_2(\mathbf{T})}^T\right) = \iota_2\left(\nabla_{q_2(\mathbf{T})}\mathcal{L}_{\mathrm{red}}\right).$$

We also use the fact that $q_2$ is a linear map whose transpose is $\iota_2$. □

*Proof of Theorem 8.* As above, let $\mathbf{R}, \mathbf{Q} = \mathtt{QR\text{-}compress}(\mathbf{A})$ be the outputs of Algorithm 1, so that $\mathbf{V} = (\mathbf{W}^{\mathrm{red}}, \mathbf{b}^{\mathrm{red}}) \in \mathrm{Param}(\mathbf{n}^{\mathrm{red}})$ is the dimensional reduction of the merged parameters $\mathbf{A} = (\mathbf{W}, \mathbf{b})$, and $\mathbf{Q} \in O(\mathbf{n}^{\mathrm{hid}})$. Set $\mathbf{T} = \mathbf{Q}^{-1} \cdot \mathbf{A} \in \mathrm{Param}^{\mathrm{int}}(\mathbf{n})$.

The action of $\mathbf{Q} \in O(\mathbf{n}^{\mathrm{hid}})$ on $\mathrm{Param}(\mathbf{n})$ is an orthogonal transformation, so the first claim follows from Lemma 22.

For the second claim, it suffices to consider the case $\eta = 1$. The general case follows similarly. We proceed by induction. The base case $k = 0$ amounts to Theorem 6. For the induction step, we set

$$\mathbf{Z}^{(k)} = \iota(\gamma_{\mathrm{red}}^k(\mathbf{V})) + \mathbf{T} - \iota(\mathbf{V}).$$

Each $\mathbf{Z}^{(k)}$ belongs to $\mathrm{Param}^{\mathrm{int}}(\mathbf{n})$, so $i_1(\mathbf{Z}^{(k)}) = \mathbf{Z}^{(k)}$. Moreover, $q_2\left(\mathbf{Z}^{(k)}\right) = \gamma_{\mathrm{red}}^k(\mathbf{V})$. We compute:

$$\gamma_{\mathrm{proj}}^{k+1}(\mathbf{Q}^{-1} \cdot \mathbf{A}) = \gamma_{\mathrm{proj}}\left(\gamma_{\mathrm{proj}}^k(\mathbf{Q}^{-1} \cdot \mathbf{A})\right)$$
$$= \gamma_{\mathrm{proj}}\left(\iota(\gamma_{\mathrm{red}}^k(\mathbf{V})) + \mathbf{T} - \iota(\mathbf{V})\right)$$
$$= \iota_1 \circ q_1\left(\iota(\gamma_{\mathrm{red}}^k(\mathbf{V})) + \mathbf{T} - \iota(\mathbf{V}) - \nabla_{\iota(\gamma_{\mathrm{red}}^k(\mathbf{V}))+\mathbf{T}-\iota(\mathbf{V})}\mathcal{L}\right)$$
$$= \iota(\gamma_{\mathrm{red}}^k(\mathbf{V})) - \iota_1 \circ q_1\left(\nabla_{\iota_1(\mathbf{Z}^{(k)})}\mathcal{L}\right) + \mathbf{T} - \iota(\mathbf{V})$$
$$= \iota(\gamma_{\mathrm{red}}^k(\mathbf{V})) - \iota_1 \circ \iota_2\left(\nabla_{q_2(\mathbf{Z}^{(k)})}\mathcal{L}_{\mathrm{red}}\right) + \mathbf{T} - \iota(\mathbf{V})$$
$$= \iota\left(\gamma_{\mathrm{red}}^k(\mathbf{V}) - \nabla_{\gamma_{\mathrm{red}}^k(\mathbf{V})}\mathcal{L}_{\mathrm{red}}\right) + \mathbf{T} - \iota(\mathbf{V})$$
$$= \iota\left(\gamma_{\mathrm{red}}^{k+1}(\mathbf{V})\right) + \mathbf{T} - \iota(\mathbf{V})$$

where the second equality uses the induction hypothesis; the third invokes the definition of $\gamma_{\mathrm{proj}}$; the fourth uses the fact that $\mathbf{Z}^{(k)} = \iota(\gamma_{\mathrm{red}}^k(\mathbf{V})) + \mathbf{T} - \iota(\mathbf{V})$ belongs to $\mathrm{Param}^{\mathrm{int}}(\mathbf{n})$; the fifth and sixth use Lemma 25 above; and the last uses the definition of $\gamma_{\mathrm{red}}$. □

## D.6 Example

We now discuss an example where projected gradient descent does not match usual gradient descent.

Let $\mathbf{n} = (1, 3, 1)$ be a widths vector. The space of parameters with this widths vector is 10-dimensional:

$$\mathsf{Param}(\mathbf{n}) = \mathrm{Hom}(\mathbb{R}^2, \mathbb{R}^3) \oplus \mathrm{Hom}(\mathbb{R}^4, \mathbb{R}) \simeq \mathbb{R}^{10}.$$

We identify a choice of parameters (in the merged notation)

$$\mathbf{A} = \left( A_1 = \begin{bmatrix} a & b \\ c & d \\ e & f \end{bmatrix} , A_2 = [g \ \ h \ \ i \ \ j] \right) \in \mathsf{Param}((1,3,1)) \tag{D.6}$$

with the point $p = (a, b, c, d, e, f, g, h, i, j)$ in $\mathbb{R}^{10}$. To be even more explicit, the weights for the first layer are $W_1 = \begin{bmatrix} b \\ d \\ f \end{bmatrix}$, the bias in the first hidden hidden layer is $b_1 = (a, c, e)$, the weights for the second layer are $W_2 = [h \ \ i \ \ j]$, and the bias for the output layer is $b_2 = g$.

The action of the orthogonal group $O(\mathbf{n}) = O(3)$ on $\mathsf{Param}(\mathbf{n}) \simeq \mathbb{R}^{10}$ can be expressed as:

$$Q \mapsto \begin{bmatrix} Q & 0 & 0 & 0 \\ 0 & Q & 0 & 0 \\ 0 & 0 & 1 & 0 \\ 0 & 0 & 0 & Q \end{bmatrix},$$

where the rows and columns are divided according to the partition $3 + 3 + 1 + 3 = 10$. Consider the function[6]:

$$\mathcal{L} : \mathsf{Param}(\mathbf{n}) \to \mathbb{R}$$
$$p = (a, b, c, d, e, f, g, h, i, j) \mapsto h(a + b) + i(c + d) + j(e + f) + g$$

By the product rule, we have:

$$\nabla_p \mathcal{L} = (h, h, i, i, j, j, 1, a + b, c + d, e + f)$$

One easily checks that $\mathcal{L}(Q \cdot p) = \mathcal{L}(p)$ and that $\nabla_{Q \cdot p} \mathcal{L} = Q \cdot \nabla_p \mathcal{L}$ for any $Q \in O(3)$.

The interpolating space is the eight-dimensional subspace of $\mathsf{Param}(\mathbf{n}) \simeq \mathbb{R}^{10}$ with $e = f = 0$ (using the notation of Equation D.6). Suppose $p' = (a, b, c, d, 0, 0, g, h, i, j)$ belongs to the interpolating space. Then the gradient is

$$\nabla_{p'} \mathcal{L} = (h, h, i, i, j, j, 1, a + b, c + d, 0)$$

which does not belong to the interpolating space. So one step of usual gradient descent, with learning rate $\eta > 0$ yields:

$$\gamma : p' = (a, b, c, d, 0, 0, g, h, i, j) \mapsto$$
$$(a - \eta h , b - \eta h , c - \eta i , d - \eta i , -\eta j , -\eta j , g - \eta , h - \eta(a + b) , i - \eta(c + d) , j)$$

On the other hand, one step of projected gradient descent yields:

$$\gamma_{\mathrm{proj}} : p' = (a, b, c, d, 0, 0, g, h, i, j) \mapsto$$
$$(a - \eta h , b - \eta h , c - \eta i , d - \eta i , 0 , 0 , g - \eta , h - \eta(a + b) , i - \eta(c + d) , j)$$

Direct computation shows that the difference between the evaluation of $\mathcal{L}$ after one step of gradient descent and the evaluation of $\mathcal{L}$ after one step of projected gradient descent is:

$$\mathcal{L}(\gamma(p')) - \mathcal{L}(\gamma_{\mathrm{proj}}(p')) = 2\eta j^2.$$

---

[6]For $\mathbf{A} \in \mathsf{Param}(\mathbf{n})$, the neural function of the neural network with affine maps determined by $\mathbf{A}$ and identity activation functions is $\mathbb{R} \to \mathbb{R}; x \mapsto \mathcal{L}(\mathbf{W})x$. The function $\mathcal{L}$ can appear as a loss function for certain batches of training data and cost function on $\mathbb{R}$.

## E  Experiments

As mentioned in Section 7, we provide an implementation of Algorithm 1 in order to (1) empirically validate that our implementation satisfies the claims of Theorems 6 and Theorem 8 and (2) quantify real-world performance. Our implementation uses a generalization of radial neural networks, which we explain presently.

### E.1  Radial neural networks with shifts

In this section, we consider radial neural networks with an extra trainable parameter in each layer that shifts the radial rescaling activation. Adding such parameters allows for more flexibility in the model, and (as shown in Theorem 26) the model compression of Theorem 6 holds for such networks. It is this generalization that we use in our experiments.

Let $h : \mathbb{R} \to \mathbb{R}$ be a function. For any $n \geq 1$ and any $t \in \mathbb{R}$, the corresponding *shifted radial rescaling function* on $\mathbb{R}^n$ is given by:

$$\rho = h^{(n,t)} : v \mapsto \frac{h(|v| - t)}{|v|} v$$

if $v \neq 0$ and $\rho(0) = 0$. A *radial neural network with shifts* consists of the following data:

1. Hyperparameters: A positive integer $L$ and a widths vector $\mathbf{n} = (n_0, n_1, n_2, \ldots, n_L)$.
2. Trainable parameters:
   (a) A choice of weights and biases $(\mathbf{W}, \mathbf{b}) \in \mathsf{Param}(\mathbf{n})$.
   (b) A vector of shifts $\mathbf{t} = (t_1, t_2, \ldots, t_L) \in \mathbb{R}^L$.
3. Activations: A tuple $\mathbf{h} = (h_1, \ldots, h_L)$ of piecewise differentiable functions $\mathbb{R} \to \mathbb{R}$. Together with the shifts, we have the shifted radial rescaling activation $\rho_i = h_i^{(n_i, t_i)} : \mathbb{R}^{n_i} \to \mathbb{R}^{n_i}$ in each layer.

The *feedforward function* of a radial neural network with shifts is defined in the usual recursive way, as in Section 3. The trainable parameters form the vector space $\mathsf{Param}(\mathbf{n}) \times \mathbb{R}^L$, and the loss function of a batch of training data $\{(x_i, y_i)\} \subset \mathbb{R}^{n_0} \times \mathbb{R}^{n_L}$ is defined as

$$\mathcal{L} : \mathsf{Param}(\mathbf{n}) \times \mathbb{R}^L \longrightarrow \mathbb{R}; \qquad (\mathbf{W}, \mathbf{t}) \mapsto \sum_j \mathcal{C}(F_{(\mathbf{W}, \mathbf{b}, \mathbf{t}, \mathbf{h})}(x_j), y_j)$$

where $F_{(\mathbf{W}, \mathbf{b}, \mathbf{t}, \mathbf{h})}$ is the feedforward function of a radial neural network with weights $\mathbf{W}$, biases $\mathbf{b}$, shifts $\mathbf{t}$, and radial rescaling activations produced from $\mathbf{h}$. We have the gradient descent map:

$$\gamma : \mathsf{Param}(\mathbf{n}) \times \mathbb{R}^L \longrightarrow \mathsf{Param}(\mathbf{n}) \times \mathbb{R}^L$$

which updates the entries of $\mathbf{W}$, $\mathbf{b}$, and $\mathbf{t}$. The group $O(\mathbf{n}^{\mathrm{hid}}) = O(n_1) \times \cdots \times O(n_{L-1})$ acts on $\mathsf{Param}(\mathbf{n})$ as usual (see Section 5.1), and on $\mathbb{R}^L$ trivially. The neural function is unchanged by this action. We conclude that the $O(\mathbf{n}^{\mathrm{hid}})$ action on $\mathsf{Param}(\mathbf{n}) \times \mathbb{R}^L$ commutes with gradient descent $\gamma$. We now state a generalization of Theorem 6 for the case of radial neural networks with shifts. We omit a proof, as it uses the same techniques as the proof of Theorem 6.

**Theorem 26.** *Let* $(\mathbf{W}, \mathbf{b}, \mathbf{t}, \mathbf{h})$ *be a radial neural network with shifts and widths vector* $\mathbf{n}$. *Let* $\mathbf{W}^{\mathrm{red}}$ *and* $\mathbf{b}^{\mathrm{red}}$ *be the weights and biases of the compressed network produced by Algorithm 1. The feedforward function of the original network* $(\mathbf{W}, \mathbf{b}, \mathbf{t}, \mathbf{h})$ *coincides with that of the compressed network* $(\mathbf{W}^{\mathrm{red}}, \mathbf{b}^{\mathrm{red}}, \mathbf{t}, \mathbf{h})$.

Theorem 8 also generalizes to the setting of radial neural networks with shifts, using projected gradient descent with respect to the subspace $\mathsf{Param}^{\mathrm{int}}(\mathbf{n}) \times \mathbb{R}^L$ of $\mathsf{Param}(\mathbf{n}) \times \mathbb{R}^L$.

 ## E.2  Implementation details

Our implementation is written in Python and uses the QR decomposition routine in NumPy [21]. We also implement a general class `RadNet` for radial neural networks using PyTorch [41]. For brevity, we write $\hat{\mathbf{W}}$ for $(\mathbf{W}, \mathbf{b})$ and $\hat{\mathbf{W}}^{\mathrm{red}}$ for $(\mathbf{W}^{\mathrm{red}}, \mathbf{b}^{\mathrm{red}})$.

**(1) Empirical verification of Theorem 6.**  We use synthetic data to learn the function $f(x) = e^{-x^2}$ with $N = 121$ samples $x_j = -3 + j/20$ for $0 \leq j < 121$. We model $f_{\hat{\mathbf{W}}}$ as a radial neural network with widths $\mathbf{n} = (1, 6, 7, 1)$ and activation the radial shifted sigmoid $h(x) = 1/(1 + e^{-x+s})$. Applying `QR-compress` gives a radial neural network $f_{\hat{\mathbf{W}}^{\mathrm{red}}}$ with widths $\mathbf{n}^{\mathrm{red}} = (1, 2, 3, 1)$. Theorem 6 implies that the neural functions of $f_{\hat{\mathbf{W}}}$ and $f_{\hat{\mathbf{W}}^{\mathrm{red}}}$ are equal. Over 10 random initializations of $\hat{\mathbf{W}}$, the mean absolute error $(1/N) \sum_j |f_{\hat{\mathbf{W}}}(x_j) - f_{\hat{\mathbf{W}}^{\mathrm{red}}}(x_j)| = 1.31 \cdot 10^{-8} \pm 4.45 \cdot 10^{-9}$. Thus $f_{\hat{\mathbf{W}}}$ and $f_{\hat{\mathbf{W}}^{\mathrm{red}}}$ agree up to machine precision.

**(2) Empirical verification of Theorem 8.**  Adopting the notation from above, the claim is that training $f_{\mathbf{Q}^{-1} \cdot \hat{\mathbf{W}}}$ with objective $\mathcal{L}$ by projected gradient descent coincides with training $f_{\hat{\mathbf{W}}^{\mathrm{red}}}$ with objective $\mathcal{L}_{\mathrm{red}}$ by usual gradient descent. We verified this on synthetic data using 3000 epochs at learning rate 0.01. Over 10 random initializations of $\hat{\mathbf{W}}$, the loss functions match up to machine precision with $|\mathcal{L} - \mathcal{L}_{\mathrm{red}}| = 4.02 \cdot 10^{-9} \pm 7.01 \cdot 10^{-9}$.

**(3) Reduced model trains faster.**  Due to the relation between projected gradient descent of the full network $\hat{\mathbf{W}}$ and gradient descent of the reduced network $\hat{\mathbf{W}}^{\mathrm{red}}$, our method may be applied before training to produce a smaller model class which *trains* faster without sacrificing accuracy. We test this hypothesis in learning the function $f : \mathbb{R}^2 \to \mathbb{R}^2$ sending $x = (t_1, t_2)$ to $(e^{-t_1^2}, e^{-t_2^2})$ using $N = 121^2$ samples $(-3 + j/20, -3 + k/20)$ for $0 \leq j, k < 121$. We model $f_{\hat{\mathbf{W}}}$ as a radial neural network with layer widths $\mathbf{n} = (2, 16, 64, 128, 16, 2)$ and activation the radial sigmoid $h(r) = 1/(1 + e^{-r})$. Applying `QR-compress` gives a radial neural network $f_{\hat{\mathbf{W}}^{\mathrm{red}}}$ with widths $\mathbf{n}^{\mathrm{red}} = (2, 3, 4, 5, 6, 2)$. We trained both models until the training loss was $\leq 0.01$. Running on a system with an Intel i5-8257U@1.40GHz and 8GB of RAM and averaged over 10 random initializations, the reduced network trained in $15.32 \pm 2.53$ seconds and the original network trained in $31.24 \pm 4.55$ seconds.

# F  Relation to radial basis function networks

In this appendix, we show that radial neural networks are equivalent to a particular class of multilayer radial basis functions networks. This class is obtained by imposing the condition that the so-called 'hidden dimension' at each layer is equal to one; the total number of layers, however, is unconstrained. To our knowledge, the literature contains no universal approximation result for this class of radial basis functions networks.

## F.1  Single layer case

We first recall the definition of a radial basis function network. A *local linear model extension of a radial basis function network* (henceforth abbreviated simply by *RBFN*) consists of:

- An input dimension $n$, an output dimension $m$, and a 'hidden' dimension $N$.

- For $i = 1, \ldots, N$, a matrix $W_i \in \mathbb{R}^{m \times n}$, a vector $b_i \in \mathbb{R}^n$, and a weight $a_i \in \mathbb{R}^m$.

- A nonlinear function[7] $\lambda : \mathbb{R} \to \mathbb{R}$.

---

[7]A more general version allows for a different nonlinear function for every $i = 1, \ldots, N$.

The feedforward function of a RBFN is defined as:

$$F : \mathbb{R}^n \to \mathbb{R}^m \qquad\qquad x \mapsto \sum_{i=1}^{N} \left(a_i + W_i(x + b_i)\right) \lambda(|x + b_i|).$$

The integer $N$ is commonly referred to as 'the hidden number of neurons'. This is a bit of a misnomer. Really there is only one layer with input dimension $n$ and output dimension $m$; the integer $N$ is part of the specification of the activation function.

We observe that if $N = 1$ and $a_1 = 0$, then the feedforward function is given by:

$$F : \mathbb{R}^n \to \mathbb{R}^m \qquad\qquad x \mapsto W\rho(x + b)$$

where $\rho$ is the radial rescaling function determined by $\lambda$. In words, one adds $b_1 = b \in \mathbb{R}^n$ to the input vector $x$, applies the activation $\rho$ to obtain new vector in $\mathbb{R}^n$, and then applies the linear transformation determined by the matrix $W_1 = W$ to obtain the output vector in $\mathbb{R}^m$. Motivated by this observation, we say that a RBFN is *constrained* if $N = 1$ and $a_1 = 0$.

## F.2   Constrained multilayer case

Next, we consider the constrained multilayer case of a radial basis functions network. Specifically, a *constrained multilayer* RBFN consists of:

- A widths vector $(n_0, \ldots, n_L)$ where $L$ is the number of layers.
- A matrix $W_\ell \in \mathbb{R}^{n_\ell \times n_{\ell-1}}$ for $\ell = 1, \ldots, L$.
- A vector $b_\ell \in \mathbb{R}^{n_\ell}$ for $\ell = 0, 1, \ldots, L - 1$.
- A nonlinear function $\lambda_\ell : \mathbb{R} \to \mathbb{R}$ for $\ell = 0, 1, \ldots, L - 1$. (Equivalently, the corresponding radial rescaling function $\rho_\ell : \mathbb{R}^{n_\ell} \to \mathbb{R}^{n_\ell}$ for $\ell = 0, \ldots, L - 1$.)

The feedforward function is defined as follows. For $\ell = 0, \ldots, L$, we recursively define $F_\ell : \mathbb{R}^{n_0} \to \mathbb{R}^{n_\ell}$ by setting $F_0(x) = x$ and

$$F_\ell(x) = W_\ell \rho_{\ell-1}(F_{\ell-1}(x) + b_{\ell-1})$$

for $\ell = 1, \ldots, L$. The feedforward function is $F_L$.

## F.3   Relation to radial neural networks

We now demonstrate that radial neural networks are equivalent to multilayer RBFNs.

**Proposition 27.** *For any radial neural network, there is a constrained multilayer RBFN with the same feedforward function. Conversely, for any constrained multiplayer RBFN, there is a radial neural network with the same feedforward function.*

*Proof.* For the first statement, let $(\mathbf{W}, \mathbf{b}, \boldsymbol{\rho})$ be a radial neural network with $L$ layers and widths vector $(n_0, \ldots, n_L)$. Recall the partial feedforward functions $G_\ell : \mathbb{R}^{n_0} \to \mathbb{R}^{n_\ell}$ defined recursively by setting $G_0(x) = x$ and

$$G_\ell(x) = \rho_\ell \left(W_\ell G_{\ell-1}(x) + b_\ell\right)$$

The feedforward function is $G_L$. Consider the constrained multilayer RBFN with $L + 1$ layers and the following:

- Widths vector $(n_0, n_1, \ldots, n_{L-1}, n_L, n_L)$. The last two layers have the same dimension.
- Weight matrices $W_\ell \in \mathbb{R}^{n_\ell \times n_{\ell-1}}$ for $\ell = 1, \ldots, L$ and $W_{L+1} = \mathrm{id}_{n_L} \in \mathbb{R}^{n_L \times n_L}$.
- A vector $b_\ell \in \mathbb{R}^{n_\ell}$ for $\ell = 1, \ldots, L$, and $b_0 = 0 \in \mathbb{R}^{n_0}$.
- A radial rescaling activation $\rho_\ell : \mathbb{R}^{n_\ell} \to \mathbb{R}^{n_\ell}$ for $\ell = 1, \ldots, L$, and $\rho_0 = \mathrm{id}_{n_0}$.

Let $F_\ell$ be the partial feedforward functions for this RBFN, defined recursively as above. We claim that

$$F_\ell(x) = W_\ell \circ G_{\ell-1}(x)$$

for any $x \in \mathbb{R}^{n_0}$ and $\ell = 1, \ldots, L$. We prove this by induction. The base case is $\ell = 1$:

$$F_1(x) = W_1 \circ \rho_0 \left( F_0(x) + b_0 \right) = W_1 x = W_1 \circ G_0(x)$$

For the induction step, take $\ell > 1$ and compute:

$$F_\ell(x) = W_\ell \circ \rho_{\ell-1} \left( F_{\ell-1}(x) + b_{\ell-1} \right) = W_\ell \circ \rho_{\ell-1} \left( W_{\ell-1} G_{\ell-2}(x) + b_{\ell-1} \right) = W_\ell \circ G_{\ell-1}(x)$$

The first claim now follows from the case $\ell = L$, using the fact that $W_{L+1}$ is the identity.

For the second statement, let $(\mathbf{W}, \mathbf{b}, \boldsymbol{\rho})$ be a constrained multilayer RBFN with $L$ layers and widths vector $(n_0, \ldots, n_L)$. Consider the radial neural network with $L + 1$ layers and the following:

- Widths vector $(n_0, n_0, n_1, \ldots, n_{L-1}, n_L)$. The first two layers have the same dimension.

- Weight matrices given by $\tilde{W}_1 = \mathrm{id}_{n_0}$ and $\tilde{W}_\ell = W_{\ell-1}$ for $\ell = 2, \ldots, L+1$.

- Bias vectors given by $\tilde{b}_\ell = b_{\ell-1}$ for $\ell = 1, 2, \ldots, L$, and $\tilde{b}_{L+1} = 0$.

- Radial rescaling activations given by $\tilde{\rho}_\ell = \rho_{\ell-1}$ for $\ell = 1, \ldots, L$, and $\tilde{\rho}_{L+1} = \mathrm{id}_{n_L}$.

One uses the recursive definition of the partial feedforward functions to show that, for $\ell = 1, \ldots, L$, we have $F_\ell(x) = W_\ell \circ G_\ell(x)$, where $F_\ell$ and $G_\ell$ are the partial feedforward functions of the RBFN and radial neural network, respectively. Then:

$$G_{L+1}(x) = \tilde{\rho}_{L+1} \left( \tilde{W}_{L+1} \circ G_L(x) + \tilde{b}_{L+1} \right) = W_L \circ G_L(x) = F_L(x),$$

so the two feedforward functions coincide. $\qquad\square$

## F.4 Conclusions

While radial neural networks are equivalent to a certain class of radial basis function network, we point out differences between our results and the standard theory of radial basis functions network. First, RBFNs generally only have two layers; we consider ones with unbounded depth. Second, to our knowledge, ours is the first universal approximation result such that:

- it uses networks in the subclass of multilayer RBFNs satisfying the constraint that all the number of 'hidden neurons' in each layer is equal to 1.

- it approximates functions with networks of bounded width.

- it can be used to approximate asymptotically affine functions, rather than functions defined on a compact domain.

Our compressibility result may apply to multilayer RBFNs where the number of 'hidden neurons' $N_\ell$ at each layer is not equal to 1, but we expect the compression to be weaker, and that constrained mulitlayer RBFNs are in some sense the most compressible type of RBFN.