# OpenReview forum: "Universal approximation and model compression for radial neural networks"
_NeurIPS.cc/2022/Conference — NeurIPS 2022 Submitted_

### Official Review · Reviewer_g15c · 2022-07-04

**Rating:** 3
**Confidence:** 3
**Soundness:** 3 good
**Presentation:** 3 good
**Contribution:** 2 fair

**Summary:**

Activation functions are a critical part of the design of neural networks. The choice of activation function in the hidden layers controls how well the network can model a training dataset. In this manuscript, the authors consider so-called radial neural networks, which have similarities to radial basis function neural networks. The authors show universal approximation in both infinite width and depth as well as present a projected gradient descent algorithm for training.

**Questions:**

- Can you better connect the NN architecture to the literature and suggest why the NN architecture is novel?

- Do the authors observe similarly training difficulties with radial neural networks as radial basis function neural networks?

- Can the authors add experiments to convince readers to use radial neural networks?

**Limitations:**

The work has little potential negative societal impact.

**Strengths And Weaknesses:**

SIMILARITY TO RBF NNs (WITH LOCAL LINEAR MODEL):
A radial basis function NN is given by (in the 2-layer case):

NN(x) = sum_{i=1}^N a_i*h(||x+b_i||)

It is sometimes convenient to extend a radial basis function NN with a local linear model to obtain (in the 2-layer case):

NN(x) = sum_{i=1}^N (a_i + w_i^T(x+b_i))*h(||x+b_i||).

This model is more general than the radial neural networks introduced in this manuscript. In particular, set a_i = 0, w_i = c_i*e_i/||x+b_i||, where c_i is a scalar and e_i is the ith unit vector to obtain:

NN(x) = sum_{i=1}^N c_i (x+b_i)/||x+b_i||*h(||x+b_i||) (= sum_{i=1}^N c_i rho(||x+b_i||)),

which I believe is an expression for 2-layer radial NN (as defined by the manuscript). Therefore, I do not find the NN architecture to be sufficiently novel to support publication.

TRAINING:
Sensitivity analysis explains the difficulties of gradient descent for radial basis function neural networks with smooth radial basis activation functions (see Karayiannis, "Gradient descent learning of radial basis neural networks." Proc. Inter. Conf. Neural Netw., IEEE, 1997). Do the authors observe similarly difficulties with radial neural networks?

EXPERIMENTS:
Separate from the novelty of the architecture and training, one would hope for experimental results that demonstrate a significant benefit over existing NN architectures (at least with regard to one example or metric).  The three experiments currently do not demonstrate any benefits but verify the theoretical contributions.

WRITING
The manuscript is well-written. Two typos are:

- p1, in (1.1): I think you want rho : R^n->R^n at the start of (1.1).
- p6, line 195: "strengthend" should read "strengthened"

---

> ### Author Response · Authors · 2022-08-01
> **Connection to RBFs; response to specfic questions**
>
> We appreciate the reviewer's careful reading of our paper and comments. While the reviewer is correct to detect a connection with radial basis functions networks (RBFNs), the relation is different than what they have written, and we believe there are factual errors in this review. In particular, our results are previously unknown in the literature.
>
> ### Connection to RBF networks
>
> As noted in the common reply, radial neural networks are special cases of *multilayer* RBFNs. We have added Appendix F in the supplementary material to clarify this point. The connection only serves to increase the relevance of our results; our findings are not subsumed by previous work.
>
> Briefly, in the notation of the review, if one takes $N=1$ and $a_1=0$, then one indeed obtains an expression that captures the feedforward functions of two-layer radial neural networks. However, rather than increase the 'hidden neurons', one expands in the depth direction to produce radial neural networks. The final equation beginning $NN(x) = \dots$ is *not* the feedforward function of a two-layer radial neural network.
>
> While universality for RBFNs has been know for a long time, the approximation is valid only on a compact domain, and depends on increasing the number $N$ of hidden neurons; in other words, expanding the width. In contrast to the standard theory, we use bounded widths to approximate asymptotically affine functions. Moreover, our universality result applies to a subclass of RBFNs, so is not automatically implied by that for all RBFNs.
>
> An analogy can be made with convolutional neural networks: the fact that convolution is a linear operation means any CNN can be implemented as a multi-layer perceptron by replacing convolution with multiplication by a Toeplitz matrix. This does not diminish the importance of CNNs; rather, they can be regarded as MLPs constrained by translation symmetries and locality assumptions.
>
> ### Typos
>
> Thanks for pointing these out. The one on page 6 is indeed a typo and we fixed it. As for the one on page 1, we reserve $\rho$ for radial rescaling functions; the function in Equation 1.1 is a pointwise activation function. Perhaps something like $\widetilde{\sigma}$ would be more appropriate, but we opted to keep new notation to a minimum.
>
> ### Literature
>
> >  Can you better connect the NN architecture to the literature and suggest why the NN architecture is novel?
>
> Radial rescaling activations have been used in many places in the literature, often in the context of equivariant neural networks (see references 26, 47, 55, 56, and 57). However, their theory contained many fundamental gaps. We fill in some of those gaps by establishing universality for neural networks with radial rescaling activations, and by providing an explicit compressibility algorithm. We note that our theoretical results rely on novel techniques that have little precedent in the literature. These include: decomposing functions in the 'depth' direction (rather than the more conventional 'width' direction), and applying QR decompositions to parameter spaces without changing the activation functions.
>
> Additionally, previous works have the activation applied along the channel dimension, not across the entire hidden layer, as we do. Previous work has  considered orthogonal equivariance only in dimensions n=2 and n=3; our orthogonal equivariance is for any dimension.
>
> Another departure from the literature is a heightened focus on  parameter space symmetries. The parameter space of every neural network has certain linear transformations that leave the loss function unchanged. Factoring out this group of symmetries can lead to compression and better convexity properties of the loss function. However, in many cases, especially when working with pointwise activations, this symmetry group is quite small (e.g., for ReLU, it contains only certain scalar multiplications). We have chosen to study radial rescaling activations in part due to the fact that the parameter space symmetry group is a product of orthogonal groups, and hence more rich than usual.
>
> ### Training
>
> > Do the authors observe similarly training difficulties with radial neural networks as radial basis function neural networks?
>
> Great question! Thanks for pointing out the paper to us. We look forward to studying this question and will update with any findings.
>
> ### Experiments
>
> > Can the authors add experiments to convince readers to use radial neural networks?
>
> We believe our paper makes a sufficient contribution to the theory of radial neural networks.  In particular, we provide theoretical support for their use and advantages versus other networks.  While comprehensive empirical studies are certainly desirable, they are beyond the scope of this work.

---

### Official Review · Reviewer_RXP7 · 2022-07-11

**Rating:** 7
**Confidence:** 4
**Soundness:** 3 good
**Presentation:** 3 good
**Contribution:** 3 good

**Summary:**

This study introduces a new deep radial neural network, shows its density in the space of `asymptotically affine' continuous functions w.r.t the sup norm, and presents a compression algorithm that has a similar taste with QR decomposition.

**Questions:**

Suggestion:
The authors introduced a new class of functions: Asymptotically affine. As pointed out in l.158, this class covers compactly supported functions. As a consequence, Theorem 3 further implies the so-called cc-universality (i.e., the density in the space of all continuous functions w.r.t. the topology of compact convergence.) I recommend the authors to clearly state that the cc-universality is a corollary of Theorem 3.


**Limitations:**

The authors have addressed the limitation in Section 8.


**Strengths And Weaknesses:**

Strength:
The proof of the main theorem is constructive as well as simple. In traditional approximation theory, the decomposition of a function into ‘width’ direction has often been investigated. That is, decomposition into coefficients and bases (or frames). On the other hand, deep learning decomposes a function into function composites, a method that has not been well investigated by traditional approximation theory. This study provides a concrete example of how to decompose a nonlinear function in the ‘depth’ direction, and I consider this is highly valuable as a result of pioneering research.

Weakness:
As the authors also pointed out in the conclusion and discussion section, the main theorem (universality) holds only for step-relu function.

---

> ### Author Response · Authors · 2022-08-01
> **Response to suggestion**
>
> Thank you for the positive assessment, and for the valuable suggestion. We have added a corollary about cc-universality.

---

### Official Review · Reviewer_nux4 · 2022-07-11

**Rating:** 6
**Confidence:** 3
**Soundness:** 2 fair
**Presentation:** 2 fair
**Contribution:** 2 fair

**Summary:**

The paper provides an analysis of radial neural networks from the viewpoint of expressivity and model compressivity.

**Questions:**

Why should we use these networks instead of the radial basis function networks, for which the theory is more understood?
Intuitively, one may expect that applying the nonlinearity on the activation norm instead of each component of the activation vector decreases expressivity. The result you provide is very generic; would it be possible to characterize the loss of expressivity for more complicated functions?
How does the number of layers/width needed to approximate a continuous function to accuracy epsilon compare to those needed for other types of architectures?
Could you please give intuition of why radial neural nets are more compressible than other architectures?
Am I right that Q in Thm 7 depends on k? This should be clarified.

**Limitations:**

I do not expect this work to have any societal impact.

**Strengths And Weaknesses:**

I have appreciated the originality of the work, namely, exploring simultaneously in a same paper the pros and cons of these architectures.
The main weaknesses is to me a lack of motivations: why should we use these networks instead of the radial basis function networks, for which the theory is more understood? After having read the paper, I am still unsure whether I should use this architecture or another, mostly for the following reason: it is unclear to me what expressivity we loose using those architectures. The authors only prove the ability of the model to represent simple continuous functions, but do not say exactly how expressivity compares to common architectures regarding state-of-the-art expressivity guarantees. Regarding paper clarity, I found the section about network compression hard to understand. The rest of the paper was well-presented. The paper is overall well written, I found few typos.

---

> ### Author Response · Authors · 2022-08-01
> **Response to specfic questions**
>
> We are grateful for the reviewer's thoughtful assessment. We address the specific questions posed in the review.
>
> ### Motivation
>
> One of our main motivations is that radial rescaling activations have appeared in many places in the literature (and are related to 'vector neurons' and other such constructions), but their theory has many fundamental gaps. We contribute to this theory by establishing universality, and by providing an explicit compressibility algorithm. Thus we demonstrate that radial rescalings lead to neural networks satisfying the necessary conditions for learning and with advantages in terms of compressibility.
>
> Another motivation is a better understanding of parameter space symmetries. The parameter space of every neural network has certain linear transformations that leave the loss function unchanged. Factoring out this group of symmetries can lead to compression and possibly better convexity properties of the loss function. However, in many cases, especially when working with pointwise activations, this symmetry group is quite small (e.g., for ReLU, it contains only certain scalar multiplications and permutations). We have chosen to study radial rescaling activations in part due to the fact that the parameter space symmetry group is a product of orthogonal groups, and hence more rich than usual.
>
> Finally, as pointed out in the common reply, our results contribute to the theory of radial basis functions networks. Radial neural networks turn out to be a certain type of radial basis function network with a high degree of symmetry and compressibility. An analogy can be made with convolutional neural networks: the fact that convolution is a linear operation means any CNN can be implimented as a multi-layer perceptron by replacing convolution with multiplication by a Toeplitz matrix. This does not diminish the importance of CNNs; rather, they can be regarded as MLPs constrained by translation symmetries and locality assumptions.
>
> We are in the process of updating the paper, and will emphasize these motivations in the introduction.
>
> ### Expressivity
>
> > The result you provide is very generic; would it be possible to characterize the loss of expressivity for more complicated functions? How does the number of layers/width needed to approximate a continuous function to accuracy epsilon compare to those needed for other types of architectures?
>
> Our universal approximation result guarantees that there is no loss of expressivity when using radial neural networks.  It is generally similar to the type of result found for other types of networks in the literature and includes several cases which are not always considered including affine asymptotic approximation and bounded width.  Please do let us know if we have misinterpreted your question.
>
> As for the second question, the answer depends on which of our universality results one uses, and which pointwise universality result one compares it to. In general, though, the widths can be much smaller than in the pointwise case, but more layers are necessary.
>
> For example, in Proposition 11 (Appendix B), we describe the number of layers needed to approximate a Lipschitz continuous function on the closed unit cube in terms of the Lipschitz constant.
>
> ### Intuition
>
> >  Could you please give intuition of why radial neural nets are more compressible than other architectures?
>
> The intuition comes down to the fact that the parameter space has a large orthogonal symmetry group, which in turn is a consequence of radial rescaling functions commuting with orthogonal transformations. The parameter space symmetries leave the feedforward function (and loss function) unchanged, so can be factored out in the appropriate sense to produce the compressed model. This 'factoring out' procedure is where the QR decomposition comes in.
>
> In more detail, for each layer $i$, the radial rescaling activation preserves the subspace of $\mathbb{R}^{n_i}$  spanned
> by the image of the linear map $W_i$ and the bias vector $b_i$ (or, proceeding by induction, their reduced versions). Hence, one can ignore elements in $\mathbb{R}^{n_i}$ not in the subspace.
>
> ### Orthogonal transformation
>
> > Am I right that Q in Thm 7 depends on k?
>
> The tuple $\mathbf{Q}$ in Theorem 7  does *not* depend on $k$. That is crucial, since it means that, at any stage of gradient descent, one can use the same orthogonal transformation to pass from the partially trained original model to the partially trained transformed model.  In particular, the pre-trained and the fully trained models are related using the same orthogonal transformation $\mathbf{Q}$. Similarly, the tuple $\mathbf{U}$ in Theorem 7 also does not depend on $k$. We updated the manuscript to emphasize this point; thanks for bringing our attention to it.

---

> ### Comment · Reviewer_RXP7 · 2022-08-06
> **What is the state-of-the-art in the study of expressivity?**
>
> Hi,
>
> I am also working on approximation theory, but have never heard of an ideas that there is a state-of-the-art on expressivity.
> Could you clarify what kind of state did you expect?

---

### Author Response · Authors · 2022-08-01
**Overall reply**

We are very thankful to each of the reviewers for their careful reading of the paper, and for their detailed comments. Reviewer nux4 points out the originality of our work: we introduce a new type of deep neural network with certain non-pointwise activation functions known as radial rescaling activations. Furthermore, as noted by Reviewer RXP7, we develop novel techniques to prove universality for these networks, as well as a model compression algorithm that can be used in conjunction with projected gradient descent. Finally, Reviewer g15c complimented the quality of the writing.

Common replies:

* The radial neural networks in our paper can be constructed by composing many 2-layer radial basis function networks (RBFNs) with local linear models and only a single neuron in each hidden RBF layer. We added Appendix F to clarify this connection and will update the introduction and conclusion sections in order to highlight this point. To our knowledge, this specific type of RBFN has not previously appeared in the literature.

* The connection with RBFNs in no way diminishes the novelty of our results; rather, it provides additional motivation and elevates their significance, as they shed new light on mulitlayer RBFNs. Specifically, to our knowledge, ours is the first bounded width universality result for such deep networks. Moreover, (1) it applies to asymptotically affine functions rather just ones defined on a compact domain and (2) we prove universality for a subclass of RBFNs, which is generally harder than proving it for the entire class. Additionally, we use previously unstudied parameter symmetries to establish compressibility properties.

*  Our work provides a foundational theoretical contribution to the study of radial neural networks, and more broadly to the understanding of non-pointwise activations. While we are very interested in empirical consequences, a full-scale empirical assessment of such networks is beyond the scope of this work.

The revisions to the paper are highlighted in blue for convenience.

---

### Meta-Review · Area_Chair_e9Ls · 2022-08-27

**Recommendation:** Reject
**Confidence:** Certain

**Metareview:**

This paper is very strong in some regards -- it gives a clear technical setup, a sophisticated but accessible analysis (impressively so), and solid coverage of related work. At the same time, I believe that it is missing key pieces -- ones that would naturally concern a reader in the NeurIPS community. The paper mainly claims a theoretical advancement, and while the path to establishing it is interesting, the end result is not especially impactful in theory, partly because it is not especially connected to other work (whether theoretical or practical).

On the side of strengths, the paper is very clear and well written, including the technical walkthough. I appreciate the intuitive support that the authors work hard to ensure (e.g. Figure 3), and the grounding in constructed examples (e.g. Figure 4). Beyond merely being useful for following, it was altogether enjoyable to read. The paper is also very thorough and complete in its presentation. There are details on every matter in the appendix.

What's missing roughly comes down to (a) grounding/motivation and (b) effect of the end result. Naturally (a) could strengthen (b), but to comment on these individually:

** Motivation

The neural net architecture studied in this paper is also first introduced in this paper. Why this architecture? It bears some relation to another architecture, namely RBFNs, but the authors stress novely of the definition. (For instance, they highlight that "this specific type of RBFN has not previously appeared in the literature.") In turn, my understanding is that the results do not directly bear on RBFNs of prior interest.

There is separately a clear opportunity to motivate radial neural nets by experiment, as highlighted by two reviewers (g15c and nux4). However the authors contend that "comprehensive empirical studies are ... beyond the scope of this work." Short of a comprehensive study, the paper does not investigate this emprically at all (the only experiments learn the scalar function exp(-x^2)), so this key question remains without even partial evidence.

** Impact of result

Say we assume the architecture is motivated, and consider the theoretical contributions. The end results are (a) a universality theorem and (b) a proof of equivalence between GD in a compressed model and PGD in an uncompressed one.

Regarding (a), I appreciate that proving universality of a strict subclass of networks is (as the author response puts it) "generally harder than proving it for the entire class". However difficulty is not an indication of impact in this case, and many types of networks are universal in some way. The bounded-width aspect makes (a) more technically substantial, and I understand that Reviewer RXP7 finds the result (and techniques) valuable in the context of approximation theory. However, I would not recommend this for NeurIPS on this grounds alone.

Regarding (b), compressibility is an appealing property, but again the impact reduces back to the question of whether the (full) model is useful (theoretically or practically). This is again where even the simplest experiment could go a long way. Better yet, a positive observation here seems like it could have really meaningful consequences!

The reviewers did not come to a consensus on ratings, but there was also no clear argument among them for acceptance. Many of the points from reviews and discussion -- both approvals and concerns -- are reflected above. Others concerns initially raised in reviews were completely and thoroughly addressed by the authors in their response. (For example, reviewer g15c initially questioned the relation to RBFNs, and the authors' reply was comprehensive.)

As a side note, the paper's font deviates from the conference format. This did not factor in my evaluation at all, but as a general tip I would avoid these sorts of modifications.

**Award:**

No

---

### Decision · Program_Chairs · 2022-09-14

Reject